# Passive Earth Observations of Volcanic Clouds in the Atmosphere

**Fred Prata** *,†(id) **and Mervyn Lynch**(id)

School of Electrical Engineering, Computing and Mathematical Sciences, Curtin University, Kent St, Bentley, Perth, WA 6102, Australia; M.Lynch@curtin.edu.au

*   Correspondence: fred@aires.space or fred_prata@hotmail.com
†   Current address: AIRES Pty. Ltd., Mt Eliza, Victoria 3930, Australia.

**Abstract:** Current Earth Observation (EO) satellites provide excellent spatial, temporal and spectral coverage for passive measurements of atmospheric volcanic emissions. Of particular value for ash detection and quantification are the geostationary satellites that now carry multispectral imagers. These instruments have multiple spectral channels spanning the visible to infrared (IR) wavelengths and provide $1 \times 1$ km$^2$ to $4 \times 4$ km$^2$ resolution data every 5–15 min, continuously. For ash detection, two channels situated near 11 and 12 μm are needed; for ash quantification a third or fourth channel also in the infrared is useful for constraining the height of the ash cloud. This work describes passive EO infrared measurements and techniques to determine volcanic cloud properties and includes examples using current methods with an emphasis on the main difficulties and ways to overcome them. A challenging aspect of using satellite data is to design algorithms that make use of the spectral, temporal (especially for geostationary sensors) and spatial information. The hyperspectral sensor AIRS is used to identify specific molecules from their spectral signatures (e.g., for SO$_2$) and retrievals are demonstrated as global, regional and hemispheric maps of AIRS column SO$_2$. This kind of information is not available on all sensors, but by combining temporal, spatial and broadband multi-spectral information from polar and geo sensors (e.g., MODIS and SEVIRI) useful insights can be made. For example, repeat coverage of a particular area using geostationary data can reveal temporal behaviour of broadband channels indicative of eruptive activity. In many instances, identifying the nature of a pixel (clear, cloud, ash etc.) is the major challenge. Sophisticated cloud detection schemes have been developed that utilise statistical measures, physical models and temporal variation to classify pixels. The state of the art on cloud detection is good, but improvements are always needed. An IR-based multispectral cloud identification scheme is described and some examples shown. The scheme is physically based but has deficiencies that can be improved during the daytime by including information from the visible channels. Physical retrieval schemes applied to ash detected pixels suffer from a lack of knowledge of some basic microphysical and optical parameters needed to run the retrieval models. In particular, there is a lack of accurate spectral refractive index information for ash particles. The size distribution of fine ash (1–63 μm, diameter) is poorly constrained and more measurements are needed, particularly for ash that is airborne. Height measurements are also lacking and a satellite-based stereoscopic height retrieval is used to illustrate the value of this information for aviation. The importance of water in volcanic clouds is discussed here and the separation of ice-rich and ash-rich portions of volcanic clouds is analysed for the first time. More work is required in trying to identify ice-coated ash particles, and it is suggested that a class of ice-rich volcanic cloud be recognized and termed a 'volcanic ice' cloud. Such clouds are frequently observed in tropical eruptions of great vertical extent (e.g., 8 km or higher) and are often not identified correctly by traditional IR methods (e.g., reverse absorption). Finally, the global, hemispheric and regional sampling of EO satellites is demonstrated for a few eruptions where the ash and SO$_2$ dispersed over large distances (1000s km).

**Keywords:** volcanic ash; Earth observations; satellites; remote sensing

## 1. Introduction

Earth Observation (EO) from orbiting satellites has a history that dates back to the 1960s. The first EO satellites were designed for measuring the Earth's atmosphere with the aim of improving weather forecasting. Around the same time, satellite instruments to study the Earth's land surface were being launched and used mainly for agricultural applications. These satellite programs: TIROS/NOAA for the atmosphere and Landsat for the land surface, continue to the present. The instruments designed for these programs had specific goals and consequently the spectral, spatial and temporal sampling strategies employed were optimised for weather and land observations. None of the instruments launched into space for EO were designed specifically for the purpose of observing volcanic clouds. For the purpose of this article, volcanic clouds are defined as any gaseous or particulate emissions from volcanoes that reach the atmosphere. The principal atmospheric volcanic gases are water vapour ($H_2O$), carbon dioxide ($CO_2$), and sulphur dioxide ($SO_2$), and the major particulate is volcanic ash, which is a class of material derived from pulverised tephra [1]. There are other gases emitted during volcanic eruptions, including hydrochloric acid (HCl) and hydrogen sulphide ($H_2S$) among others, but these occur in much lower quantities during most volcanic eruptions. $CO_2$ and $H_2O$ are ubiquitous in the atmosphere and it has been difficult to measure these volcanically generated gases from the background concentrations. Thus for at least the past 40 years the main volcanic focus of atmospheric EO has been on detecting and quantifying $SO_2$ and volcanic ash. As will be shown, both $SO_2$ and volcanic ash can be detected, and in many cases quantified, by EO sensors not specifically designed to measure them. Both $SO_2$ and volcanic ash have important impacts on the atmosphere: $SO_2$ is converted to sulfuric acid to form small droplets that affect the Earth's radiation balance by reflecting solar radiation away from the surface [2]. Volcanic ash can also affect the radiation balance, but its lifetime in the atmosphere is short (typically a few days) so its effects are mostly local. However, volcanic ash can damage jet engines when dispersing ash clouds intersect commercial jet airline flight routes. The potential hazard is sufficient to inflict economic damage [3,4].

EO sensors are now used routinely by operational agencies to monitor volcanic clouds and quantify their mass loadings to provide warnings for commercial jets. There are now, in fact, a large number of EO satellites carrying sensors capable of detecting and quantifying volcanic clouds, so many that a new era of research into volcanic clouds has begun. The eruption of Eyjafjallajökul in April and May 2010 brought the volcanic ash hazard to the attention of the flying public and more importantly to airlines, regulators, meteorological offices, research institutes and funding agencies. The need to improve observation and forecasting of volcanic clouds has driven new research and brought some significant innovations to address the problems. It is appropriate then to review, explore and summarise existing volcanic cloud research using EO sensors and explore new research possibilities driven by EO sensor capabilities in multi-spectral, multi-temporal and multi-angle sensors as well as exploit higher spatial resolution sensors, which have in the past been less studied.

This article begins with some summary material on the satellites and sensors either routinely used for volcanic cloud studies or possessing capabilities to generate new information on them. Following this, a review of the EO methods used to detect and quantify volcanic ash and $SO_2$ gas is provided. The emphasis is on volcanic ash as this tends to be more difficult to detect and measure, and the fundamental ideas surrounding infrared (IR) detection are presented, without going into too much detail on radiative transfer. Methods of detection are prioritised over quantification and retrieval as there is still much room for improvement in this area. An example of a multi-spectral cloud classification scheme is given. In the methods section, details of schemes to exploit spectral and temporal information are described and an example of the use of recent Sentinel–3 multiangle data to determine cloud altitude is given. A newly recognised type of atmospheric hydrometeor, "volcanic ice"

is defined and discussed and some possible approaches for its detection are suggested. Finally, global maps of ash and $SO_2$ are included based on EO multispectral sensor retrievals to show how EO data can be used to assess the annual, global input of volcanic emissions to the atmosphere.

## 2. Satellite Orbits and Sensors

### 2.1. Polar Orbits

The polar orbit describes a low-Earth orbit that travels from pole to pole at a high inclination angle, typically ∼98°. These orbits are often arranged to be Sun-synchronous; an arrangement that causes the satellite to cross Earth's equatorial plane at the same local time each day. In order for this to happen the orbit must precess by ∼1° E each day. The plane of the orbit then slowly rotates with respect to Earth's axis. The purpose of a Sun-synchronous orbit is to provide similar solar illumination of targets each day for optical sensors. Examples of satellites that are in Sun-synchronous orbits and their principal orbital parameters are provided in Table 1.

**Table 1.** Non-exhaustive list of Earth obervation (EO) near polar-orbiting satellites carrying instruments useful for volcano monitoring from the 1970s to present. There are many other satellite platforms but this table contains the most commonly used. A = Ascending node; D = Descending node. See [5] for an encyclopaedic description of satellites and systems for observing the Earth.

| Satellite | Local Equatorial Crossing Time | Inclination (Degrees) | Height (km) | Period (Minutes) | Repeat Cycle (Days) |
|---|---|---|---|---|---|
| Landsat-5 | 09:45 | 98.2 | 704 | 99 | 16 |
| Landsat-7 | 10:00 | 98.2 | 705 | 99 | 16 |
| Landsat-8 | 10:30 | 98.2 | 701–703 | 98.8 | 16 |
| NOAA-11 | 13:40(A) | 98.9 | 845–863 | 102.1 | 11 |
| NOAA-12 | 19:30(A) | 98.7 | 806–825 | 101.3 | 11 |
| NOAA-13 | | Failed 11 days after launch | | | |
| NOAA-14 | 13:40(A) | 98.9 | 848–861 | 102.1 | 11 |
| NOAA-15 | 16:44(A) | 98.7 | 804–818 | 101.3 | 11 |
| NOAA-16 | 14:00(A) | 98.74 | 845–860 | 102.1 | 11 |
| NOAA-17 | 22:00(A) | 98.52 | 800–817 | 101.1 | 11 |
| NOAA-18 | 14:00(A) | 99.1 | 840–862 | 102 | 11 |
| NOAA-19 | 13:34(A) | 99.1 | 840–862 | 102 | 11 |
| NPP | 13:30(A) | 98.74 | 824 | 101 | 16 |
| ERS-1 | 10:30(D) | 98.52 | 782–785 | 100 | 35 |
| ERS-2 | 10:30(A) | 98.5 | 780 | 100 | 35 |
| ENVISAT | 10:30(A) | 98.5 | 780 | 100 | 35 |
| Aqua | 13:30(A) | 98.2 | 705 | 98.8 | 16 |
| Terra | 10:30(D) | 98.5 | 705 | 99.0 | 16 |
| Aura | 13:45(A) | 98.7 | 705 | 98.8 | 16 |
| MetOP-A/B/C | 21:30 (A) | 98.7 | 817–827 | 101 | 29 |
| Sentinel-2A/2B | 10:30 | 98.62 | 786 | 100.6 | 10 |
| Sentinel-3A/3B | 10:00 | 98.65 | 814.5 | 100.99 | 27 |

### 2.2. Geostationary Orbits

A geostationary satellite orbits the Earth with the same period as the period of rotation of the Earth. This permits the satellite to view the same region of the Earth continuously. Orbital mechanics constrains this orbit to be over the equator at a distance of approximately six Earth radii from the Earth's centre or ∼36,000 km above the Earth's surface. The satellite can be placed at any longitude. The great distance from the Earth's surface means low spatial resolution unless a large telescope is employed. The great advantage of these satellites is that they can provide very high (a few minutes) temporally resolved data and this is a natural advantage for observing short-lived, sporadic and unpredictable events like volcanic activity, except at high latitudes of both hemispheres. Figure 1a–e shows some example imagery from five of the meteorological geostationary platforms and an image

acquired from the EPIC sensor on the DSCOVR platform (Figure 1f). DSCOVR is located at the L1 Lagrange point between the Earth and the Sun at ~1.5 million km from the Earth. In this location the platform is in a semi-stable position and observes the sunlit side of the Earth continuously as it rotates beneath the satellite. An advantage of this position for observing volcanic emissions is that the EPIC UV sensor can utilise sunlight to measure $SO_2$ columns continuously [6,7].

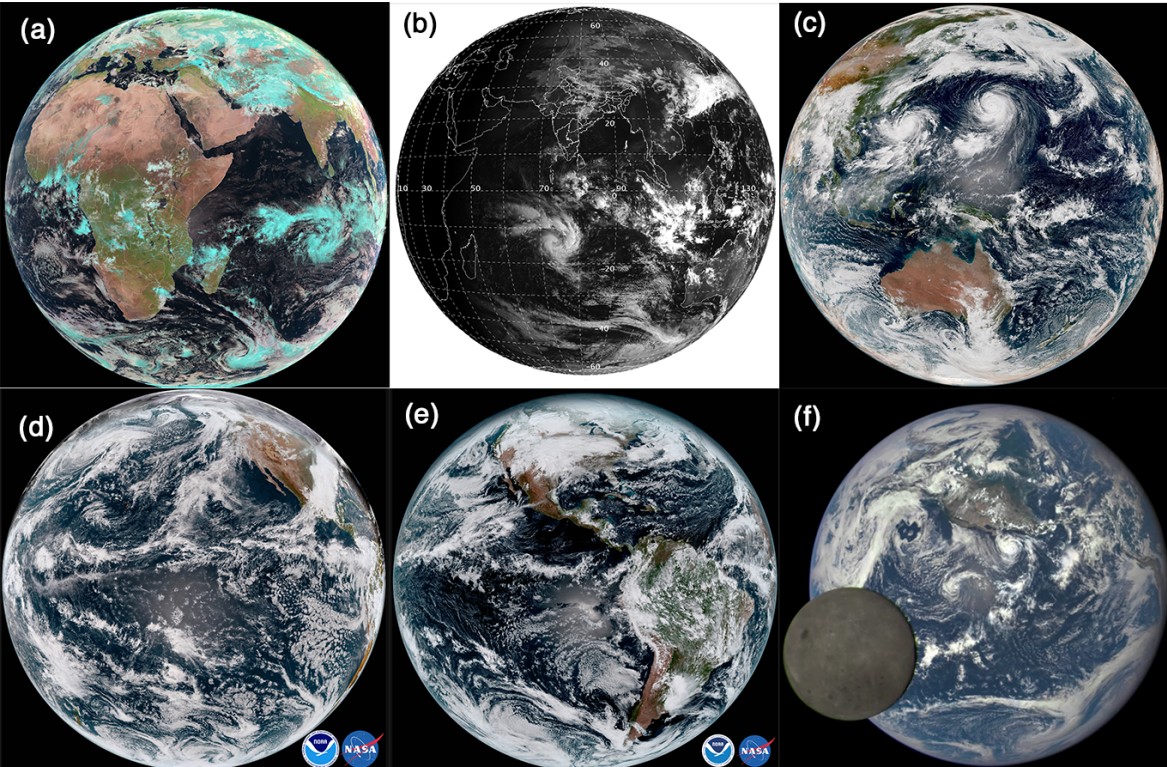

**Figure 1.** Earth observing geostationary satellite platforms. (**a**) MSG-3 at 0° longitude (true-colour image). (**b**) INSAT at 82° E longitude (single band visible). (**c**) Himawari-8 at 140° E longitude (true-colour image). (**d**) GOES-17 at 137.2° W longitude (true-colour image). (**e**) GOES-16 at 0° longitude (true-colour image). (**f**) EPIC/DSCOVR at the L1 lagrange point (true-colour) also showing the far side of the moon during a transit. Data courtesy of the meteorological agencies: Eumetsat (EU), NMSC (Government of India), JMA/JAXA (Japan), NASA and NOAA (USA).

*2.3. Sensors*

EO sensors useful for volcanic ash and gas measurements span the entire electromagnetic spectrum from the UV to the microwave. The discussion here is restricted to passive sensing, although active light (lidar) [8] and microwave (radar) [9] and radiometers [10] are becoming increasingly important for measuring aspects of volcanic emissions that are difficult from passive data (e.g., height and large particles in the size distribution). It is not possible to provide a comprehensive treatment of all of the sensors available for volcano measurements and we concentrate only on the most common and those that have an infrared capability, which allows for continuous 24 h monitoring. Suffice to say, in the infrared the main regions used for measurements of both gases and ash particles lie between 3–4 μm and 7–14 μm. The region between 8–12 μm is most useful because ash particles exhibit dispersive behaviour there and $SO_2$ gas has a strong absorption band around 8.6 μm. The mid-infrared region (3–4 μm) has been found to provide useful information on volcanic activity because of the heat signature from the molten rock [11,12]. There is also a weak $SO_2$ absorption situated close to 4 μm, but this can only be exploited in the case of strong emissions that are near to the surface. There is also a challenge with using the 3–4 μm region because the signal derives from both thermal emission and solar reflection and so the two components must be disentangled.

The ultra-violet region (280–340 nm) has been used extensively for measurements of volcanic $SO_2$ from the TOMS sensor [13], OMI [14], GOME and GOME-2 [15], SCHIAMACHY [16], OMPS [17] EPIC/DSCOVR [7] and Tropomi [18]. All of these sensors rely on the absorption and scattering of UV light by $SO_2$; details of these sensors can be found in the references cited.

The sensors of interest here are the ones that include infrared bands (see Table 2) the methods used to retrieve ash properties from them are described next.

**Table 2.** Summary of the important aspects of polar (p) and geostationary (g) sensors used for $SO_2$ gas and ash measurements. Note that for some of the heritage instruments (e.g., AVHRR, along-track scanning radiometer (ATSR), ans HIRS the detailed wavelength specifications may be slightly different. The sampling time for the geostationary sensors is provided in brackets, in minutes. The period of operation is given as year of launch to month and year of decommissioning. Further details of the sensors and their platforms can be found at: https://www.wmo-sat.info/oscar/instruments

| Sensor | Ash Bands μm | SO₂ Bands μm | Resolution km | Platform pol or geo | Time Period Years |
|---|---|---|---|---|---|
| AVHRR-2/3 | 3.7, 10.8, 12.0 | – | 1 | p | 1979–present |
| HIRS-2/3 | 3.76–4.57, 11.11, 12.47 | 7.3, 8.2 | 26 × 42 | p | 1979-present |
| MODIS | 3.75–4.5, 8.6, 11.03, 12.03 | 7.33, 8.55 | 1 | p | 2000–present |
| SEVIRI | 3. 8.7, 10.8, 12.0 | 7.35, 8.7 | 2 | g (15) | 2004–present |
| IMAGER/MTSAT-2 | 3.75, 10.8, 12.0 | – | 4 | g (30) | 2006–05/2016 |
| AHI/HIMAWARI-8 | 3.85, 8.60, 10.4, 11.2, 12.4 | 7.35, 8.6 | 2 | g (10) | 2004–present |
| ABI | 3.9, 8.5, 10.2, 11.2, 12.3 | 8.5 | 2 | g (15) | 2017–present |
| AIRS | 3.74–4.61, 8.80–15.4 | 6.2–8.22 | 13.5 | p | 2002-present |
| IASI | 3.62–5.00, 8.26–15.50 | 5–8.26 | 12.0 | p | 2007–present |
| ASTER | 8.30, 8.65, 10.6, 11.3 | 8.30, 8.65 | 0.09 | p | 2000–present |
| ATSR/ATSR-2/AATSR | 3.7, 10.85, 12.0 | – | 1 | p | 1991–03/2000 |
| SLSTR | 3.74, 10.85, 12.0 | – | 1 | p | 07/2016–present |
| TM/Landsat-5 | 11.45 | – | 0.12 | p | 1984–06/2013 |
| ETM+/Landsat-7 | 11.45 | – | 0.06 | p | 1999–present |
| TM/Landsat-8 | 10.8, 12.0 | – | 0.1 | p | 1982-11/2011 |

## 3. Methods–Volcanic Ash Detection and Retrieval

### 3.1. Physical Principles of Ash Detection in the Infrared

Volcanic ash is a hazard to aircraft [3,19,20]. The problem of detecting volcanic clouds from satellites is really a problem of discrimination. Clouds absorb, emit and scatter radiation in the visible, infrared and microwave regions of the electromagnetic spectrum. At visible wavelengths, depending on the geometry of illumination (by the Sun or using a laser light source) and the geometry of observation, clouds may appear bright or dark. This is true of clouds of water, ice, silicates (volcanic ash), wind blown dust (desert dust), smoke (e.g., from a large forest fire) or any other naturally or anthropogenically generated cloud of particles. It is sometimes very clear that a particular cloud is meteorological in origin (for example, a cloud of water droplets or ice particles, or a mixed phase cloud), but often not so clear that it is not a meteorological cloud. By using objective analysis of daytime visible imagery alone, it has been very difficult to unambiguously discriminate ash clouds from other clouds. During the nighttime, the task is made even more difficult. This is the main reason why researchers have turned their attention to using infrared data [21–25]. The foundations of detecting and quantifying volcanic ash in the atmosphere based on Earth observations require a good understanding of the main radiative processes affecting the measurements. In the following subsections we outline the radiative transfer theory needed and illustrate how to solve various problems depending on what information is available. Inversion methods are described and some simple examples are provided. An alternate method of determining the size distribution based on a formulation for the optical depth is proposed. We appeal to a simple model to show the effects of optical properties of particles, interfering substances

(e.g., water vapour) and describe a cloud detection scheme aimed at reducing false detections of volcanic ash clouds.

### 3.2. Modelling Radiative Transfer in Ash Clouds

The radiatve tranfser for infrared radiation passing through the atmosphere and interacting with clouds and the surface is illustrated in Figure 2.

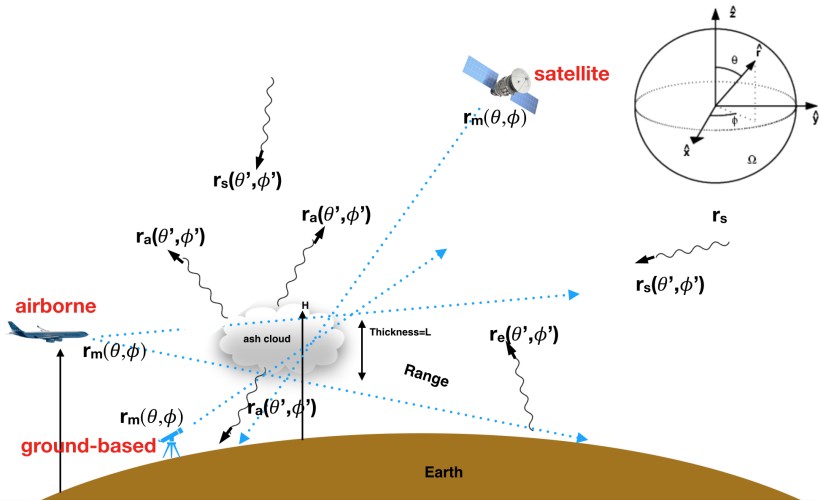

**Figure 2.** Radiative transfer problem. The inset plot (top-right) shows the definitions of the angles $\theta$ and $\phi$.

The radiation reaching satellite, the measured radiance ($r_m$) may be split into three components:

$$r_m(\theta, \phi) = r_e(\theta, \phi) + r_a(\theta, \phi) + r_s(\theta, \phi), \tag{1}$$

where $r_e$ is the radiance from the Earth (surface or ocean), $r_a$ is the radiance from the atmosphere, which can include a cloud or clouds, and $r_s$ the radiance from space, the region outside the atmosphere. The last term is negligible for nearly all problems encountered in determining cloud properties. It is understood that all the radiances are wavelength dependent and that the measurement is made over a small solid angle. The first term is the emission from the atmosphere and may be written,

$$r_a = \int_{p_s}^{0} J_\lambda(p) \frac{d\tau_\lambda(\theta, \phi)}{dp} dp, \tag{2}$$

where $p$ is pressure, $p_s$ is surface pressure, $\lambda$ is wavelength and $J$ is the source function. The source function can be quite complicated but can be represented by the Planck function if scattering is ignored,

$$J_\lambda(p) = B_\lambda[T_p].$$

The second term is the emission from the surface and may be written,

$$r_e(\theta, \phi) = \tau_s(\theta, \phi) \left\{ \epsilon_{11} \lambda(\theta, \phi) B_\lambda[T_s] + \int \int r_a \downarrow (\theta', \phi') \rho_\lambda(\theta', \phi') \sin \theta' \cos \theta' d\theta' d\phi' \right\}, \tag{3}$$

where $\tau_s$ is the transmittance (surface to atmosphere), $\epsilon$ is the surface emissivity, $B_\lambda$ is the Planck function, $T_s$ is the surface temperature, $r_a \downarrow$ is the downwelling emission from the atmosphere and

$\rho_\lambda$ is the surface bidirectional reflectance distribution function (BRDF). Further simplifications can be made if it is assumed that the surface is Lambertian and isotropic, then:

$$\rho_\lambda = \frac{1}{\pi}(1 - \epsilon_\lambda).$$

Putting this all together the forward problem may be written,

$$r_m = \tau_s(\theta, \phi) \left\{ \epsilon_{11}\lambda(\theta, \phi)B_\lambda[T_s] + \frac{1}{\pi}(1 - \epsilon_\lambda)r_a \downarrow \right\} + \int_{p_s}^{0} J_\lambda(p)\frac{d\tau_\lambda(\theta, \phi)}{dp}dp. \tag{4}$$

This equation describes the components of the radiation that reach the satellite instrument and constitute the measurement. In practice the goal is to determine some aspect of the atmosphere, such as cloud properties or the temperature change with altitude and it is the inverse problem that must be tackled. This can be stated as,

$$\mathbf{r} = \mathbf{F}(\mathbf{x}) + \mathbf{e}, \tag{5}$$

where $\mathbf{r}$ is the measurement (a vector), $\mathbf{F}(\mathbf{x})$ is the forward model and $\mathbf{e}$ is an error. $\mathbf{x}$ is the state vector which describes the physical parameters of the model, e.g., optical depth, temperature profile, cloud height, thickness, size distribution and so on. Equation (5) is a nonlinear inverse problem and its solution is sought through linearisation around a mean state $\mathbf{x_o}$:

$$\mathbf{r} - \mathbf{r_o} = \frac{\partial \mathbf{F}(\mathbf{x})}{\partial \mathbf{x}}(\mathbf{x} - \mathbf{x_o}) + \mathbf{e}, \tag{6}$$

where $\mathbf{r_o}$ is a measurement corresponding to the mean state—this can be based on a priori knowledge or a climatology. There are several excellent references and texts [26,27] on how to tackle the inverse problem, noting the many pitfalls and issues with under-determined systems of equations. Here we state the inverse problem in a form that has become familiar in atmospheric problems [26].

$$\hat{\mathbf{r}} = \mathbf{r_p} + \mathbf{S_p}\mathbf{K}^\mathbf{T}(\mathbf{K}\mathbf{S_p}\mathbf{K}^\mathbf{T} + \mathbf{S_e})^{-1}(\mathbf{y} - \mathbf{K}\mathbf{r_p}), \tag{7}$$

where $\hat{\mathbf{r}}$ is the estimate of the measurement, $\mathbf{r}_p$ is a prior estimate, $\mathbf{S}_p$ is the a priori error covariance matrix, $\mathbf{K}$ is a matrix of Frechet derivatives, $\mathbf{S}_e$ is the measurement covariance matrix and $\mathbf{y}$ is the forward model. Specific examples of methods of solving (7) can be found in [28–30].

To illustrate the ideas expressed above, consider the highly simplified problem shown in Figure 3. The number of unknowns is $2 + n$, where $n$ is the number of cloud emissivities. So the problem can only be solved after using an assumption or adding in a new measurement.

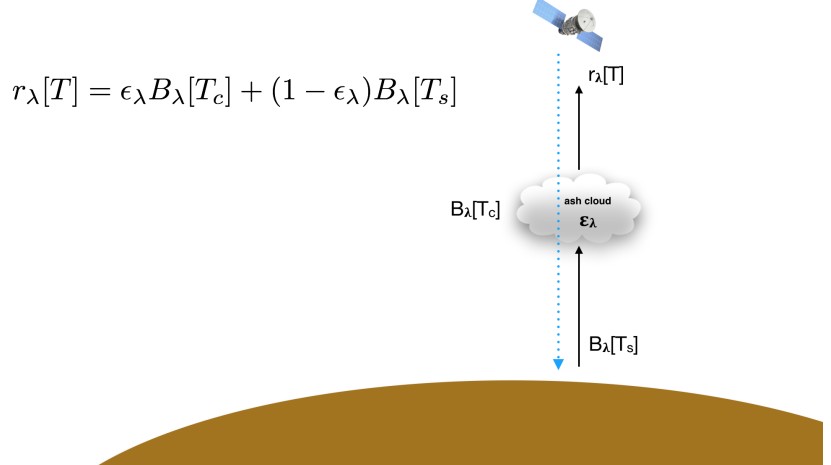

$$r_\lambda[T] = \epsilon_\lambda B_\lambda[T_c] + (1 - \epsilon_\lambda)B_\lambda[T_s]$$

**Figure 3.** Radiative transfer problem for a cloud embedded in a transparent atmosphere and viewed directly from above by a satellite-borne infrared instrument.

Assume that $T_c$ and $T_s$, the cloud top and surface temperatures respectively, are known (or measured some other way), then using Equation (7) with just two measurements, after linearisation,

$$\begin{bmatrix} \delta r_1 \\ \delta r_2 \end{bmatrix} = \begin{bmatrix} \frac{\partial r_1}{\partial \epsilon_1} & 0 \\ 0 & \frac{\partial r_2}{\partial \epsilon_2} \end{bmatrix} \begin{bmatrix} \delta \epsilon_1 \\ \delta \epsilon_2 \end{bmatrix} \tag{8}$$

$$\delta r_i = r_i - r_i^o$$

$$\delta \epsilon_i = \epsilon_i - \epsilon_i^o$$

$$\begin{bmatrix} \delta \epsilon_1 \\ \delta \epsilon_2 \end{bmatrix} = \frac{1}{\left(\frac{\partial r_2}{\partial \epsilon_2}\right)\left(\frac{\partial r_1}{\partial \epsilon_1}\right)} \begin{bmatrix} \frac{\partial r_2}{\partial \epsilon_2} & 0 \\ 0 & \frac{\partial r_1}{\partial \epsilon_1} \end{bmatrix} \begin{bmatrix} \delta r_1 \\ \delta r_2 \end{bmatrix}. \tag{9}$$

The solution is:

$$\delta \epsilon_i = \frac{B_i[T - i] - B_i[T_o]}{B_i[T_c] - B_i[T_s]}. \tag{10}$$

Quite often it is problematic to obtain independent measurements or impose a priori constraints. A strategy that can be used for volcanic cloud sensing with IR imagery is to use nearby data and assume some level of homogeneity. For example in the case of two channel data, nearby overcast and clear-sky pixels can be found and used as extra measurements in the inversion process. Assuming that $r_3 = B_1[T_c]$ and $r_4 = B_i[T_s]$, the problem becomes:

$$\begin{bmatrix} \delta r_1 \\ \delta r_2 \\ \delta r_3 \\ \delta r_4 \end{bmatrix} = \begin{bmatrix} \frac{\partial r_1}{\partial \epsilon_1} & 0 & \frac{\partial r_1}{\partial T_c} & \frac{\partial r_1}{\partial T_s} \\ 0 & \frac{\partial r_2}{\partial \epsilon_2} & \frac{\partial r_2}{\partial T_c} & \frac{\partial r_2}{\partial T_s} \\ 0 & 0 & 1 & 0 \\ 0 & 0 & 0 & 1 \end{bmatrix} \begin{bmatrix} \delta \epsilon_1 \\ \delta \epsilon_2 \\ B_1[T_c] \\ B_1[T_s] \end{bmatrix}. \tag{11}$$

Since the inverse exists, this problem can be solved analytically. There are other strategies than can be adopted to constrain the problem and these are discussed later. In the next sections the effects of optical properties and of a non-transparent atmosphere are investigated.

### 3.3. Heuristic Model

A simple model [31] based on two infrared channels is used to illustrate the effects of optical properties on the detectability and quantification of volcanic ash. The heuristic model is,

$$\Delta T = \Delta T_c (X - X^\beta), \tag{12}$$

$$X = 1 - \frac{\Delta T_1}{\Delta T_c}, \tag{13}$$

$$\epsilon_i = 1 - \exp(-k_i z), \tag{14}$$

$$\beta = \frac{k_2}{k_1}, \tag{15}$$

where $k_i$ is the absorption coefficient of the particles at wavelength $i$ and $z$ is the geometric thickness of the cloud, $\Delta T = T_1 - T_2$, $\Delta T_c = T_s - T_c$, and $\Delta T_1 = T_s - T_1$. The important physics is captured in the parameter $\beta$, the ratio of extinction coefficients at two wavelengths, $\lambda_1$ and $\lambda_2$. It is possible to explore the parameter range of $\beta$, guided by typical values for the extinction coefficients. Since the two wavelengths are usually close together the extinction coefficients are not very different and $\beta \approx 1$. When $\beta < 1$, the extinction coefficient at $k_{\lambda_1} > k_{\lambda_2}$, which leads to a "U-shaped" curve, while for the opposite case $\beta > 1$ an "arch-shaped" curve results. If $\beta = 1$ then $\Delta T$ has no variation with $T_1$ and no information can be retrieved from this analysis. It turns out that for $\lambda_1 \approx 11$ µm and $\lambda_2 \approx 12$ µm, typical of channels on many satellite sensors, $\beta < 1$ for silicates (ash particles) and $\beta > 1$ for water molecules and ice particles. If one were free to design a sensor solely for ash detection, it would be sensible to select channels that optimise the "U-shaped" curve.

### 3.4. Solving the Heuristic Model

The simple model proposed here can be solved to determine the optical depth and radius corresponding to two measurements ($T_{11}$, $T_{12}$). Figure 4 shows curves generated from the heuristic model that illustrate the effects of particle radius and optical depth (infrared opacity) on the brightness temperatures and their difference. Each curve corresponds to a different mean effective radius (indicated in red), and optical depth (indicated in green).

The scatter of black points are actual SEVIRI observations, truncated with a value of brightness temperature difference ($BTD$) = 0 K. By interpolating the curves in the data space ($T_4$, $BTD = T_4 - T_5$) (We use the subscripts 4 and 5 to indicate channels with wavelengths centred at 11 µm and 12 µm, a legacy from the use of AVHRR data, and more generally subscripts 1 and 2 to indicate two different channels, where channel 1 has a central wavelength *smaller* than channel 2), values of the mean effective radius and infrared optical depth can be determined. In this example solutions are shown for points 1 and 2. It can be seen that as $BTD$ approaches 0, multiple solutions for the optical depth are realised for a single value of the effective radius. $BTD$ can approach 0 when either the ash clouds are very thick (then $T_4$ approaches $T_c$, the cloud-top temperature) or when the ash clouds are thin (then $T_4$ approaches $T_s$, the surface temperature).

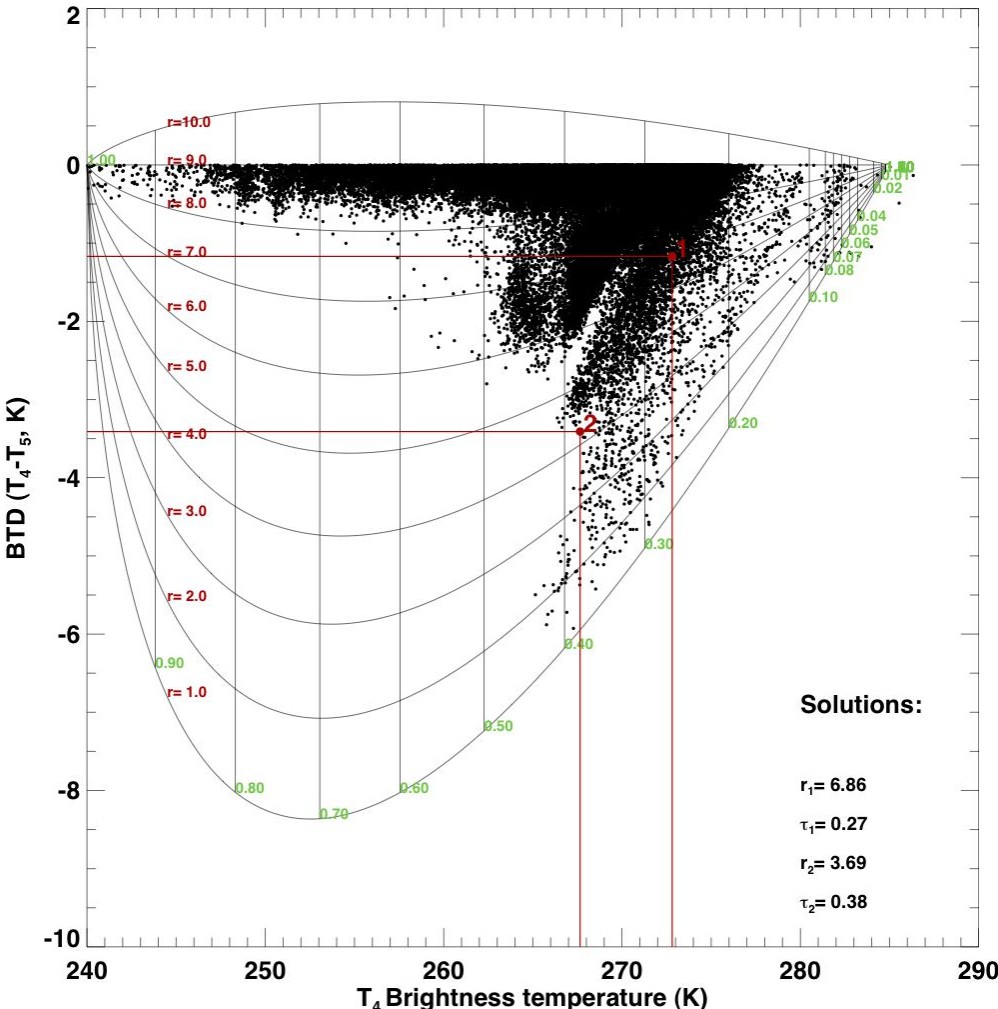

**Figure 4.** Illustration of the relation between the retrieved quantities *r*, the effective particle radius, $\tau$ the optical depth and brightness temperatures. The black dots are data points. The nearly vertical lines are isolines of optical depth (green) and the curves correspond to different effective radii.

### 3.5. Correcting for Water Vapour Effects

One of the main problems with identifying ash in a cloud arises because often the ash is in a mixture with water molecules or ice particles. Water and ice clouds have $\beta > 1$ and therefore cause an opposite effect to that caused by ash clouds on the *BTD* vs. $T_1$ diagram. The simple model can be examined further to correct for water vapour effects, or at least to understand how these effects manifest themselves. Assuming that the temperature difference observed arises from a linear combination of the signal from ash ($\Delta T_{ash}$) and the signal due to water vapour ($\Delta T_{wv}$). If the fraction of ash in the mixture is *F*, then we may write the observed temperature difference as,

$$\Delta T = F\Delta T_s[Z - Z^{\beta}], \tag{16}$$

where,

$$Z = 1 - \frac{1}{F}\frac{\Delta T_1}{\Delta T_s}.$$

The other parameters are defined as before. Ideally the water vapour correction procedure needs to be automated. There are several parameters that can be determined from the image data. These are:

(1)   the clear-sky surface temperature $T_s$,

(2)　the cloud-top temperature $T_c$,
(3)　the clear-sky value of the water vapour correction, and
(4)　the ratio of extinction coefficients $\beta$ that governs the magnitude of the "U-shaped" distribution of negative differences.

A procedure for estimating these parameters from image data has been developed. A brief outline is given below.

1.　$T_s$. This is easily estimated by finding the maximum value of $T_1$ occurring in the data.
2.　$T_c$. This is more difficult to estimate from the data, because the lowest value may not necessarily correspond to the volcanic cloud. However, provided an area in close proximity to the suspect cloud can be delineated it may be reasonable to assume that the lowest value is the cloud-top temperature.
3.　Water vapour correction. An empirical relation [32] between the precipitable water in an atmospheric column and the brightness temperature difference ($T_1 - T_2$) is used to estimate the water vapour effect

$$\Delta T_{wv} = \exp\left[6T_* - b\right], \tag{17}$$

where $T_* = T_1/T_{max}$, and $T_{max}$ is an arbitrary normalisation constant assigned a value of 320 K. The free parameter $b$ essentially determines the value of the water vapour effect on $T_1 - T_2$ at the maximum value of $T_1$. Hence $b$ can be determined directly from the image data, allowing realistic flexibility on the size of the water vapour correction determined by this semi-empirical approach.
4.　$\beta$. Theoretical estimates of $\beta$ suggest a value of around 0.7. A method for estimating $\beta$, $T_s$ and $T_c$ simultaneously has been developed by using the distribution of $T_1$ vs. $T_1 - T_2$. The distribution is first histogrammed (or binned) into intervals of 0.5 K in $T_1$. Then, the lowest values in each bin are found and a curve is generated giving the outline of the distribution. The curve is smoothed and fitted using a nonlinear least squares model. The model has three parameters, viz.: $T_s$, $T_s - T_c$ and $\beta$ that can be estimated from the fit.

The curve-fitting procedure uses the model developed earlier with $F = 1$ and the partial derivatives of the model, which are analytic. Writing,

$$Y = \alpha(X - X^\beta),$$

where, $Y = T_1 - T_1$, $\alpha = T_s - T_c$, $X = 1 - \frac{\gamma}{\alpha}$, $\gamma = T_s - T_1$. The partial derivatives are,

$$\frac{\partial Y}{\partial \alpha} = (X - X^\beta) + \frac{\gamma}{\alpha} + \beta\frac{\gamma}{\alpha}(1 - \frac{\gamma}{\alpha})^{\beta-1},$$

$$\frac{\partial Y}{\partial \beta} = \alpha X^\beta Log\beta,$$

$$\frac{\partial Y}{\partial \gamma} = 1 - \beta X^{\beta-1}.$$

An example of the correction procedure is shown in Figure 5. The characteristic " U-shaped" curve indicating ash is apparent in the uncorrected (black dots) and water vapour-corrected (red dots) data. The solid line is determined from the simple model using a suitable value of $\beta$ and values for $T_s$ and $T_c$. The important point to note is that the water vapour correction does not simply decrease all the values uniformly, rather the correction rotates the points in a clockwise direction about a point close to $\Delta T = 0$ and $T_1 = T_s$. This gives larger correction to points closer to $T_s$; points that are nearer the surface and hence expected to be affected greater by water vapour.

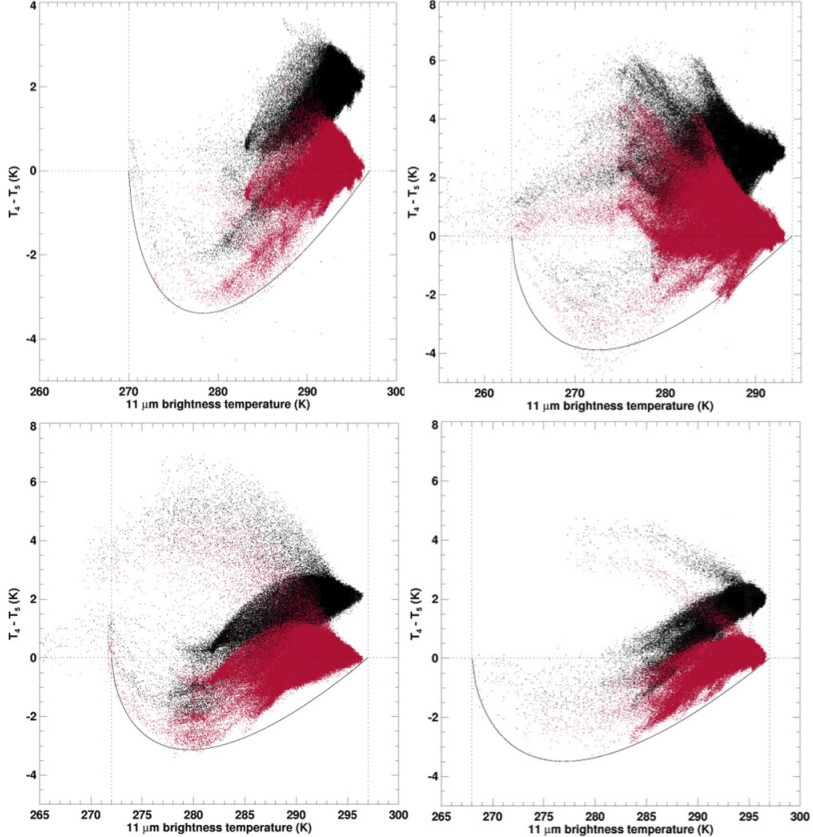

**Figure 5.** Temperature difference distributions without (black dots) and with (red dots) a water vapour correction for four different volcanic clouds using AVHRR data. $T_4$ and $T_5$ are AVHRR channel 4 (11 μm) and 5 (12 μm) brightness temperatures and correspond to $T_1$ and $T_2$ nomenclature used here.

The simple model considers only absorption as the process for extinction of IR radiation. In reality scattering is also important and unfortunately this requires more complex modelling and the use of numerical methods.

## 4. Complex Radiative Transfer Model

Several radiative transfer (RT) models have been proposed to solve the IR absorption/scattering processes for a volcanic ash cloud [22,28,29,31,33–39]. The models essentially follow the theory outlined above, but the methods of solution differ in detail.

Retrieval schemes [31,34] include a microphysical model of the ash particles with a detailed radiative transfer model, to invert the infrared data to reveal an effective particle size, cloud opacity and mass loading. When these parameters are integrated over the area covered by the cloud, the total mass can be inferred from the data. These are quantifiable products that may be incorporated with dispersion models to generate risk maps for use by the aviation industry. An example of this kind of retrieval is given in Figure 6, for the June, 2011 eruption of Puyehue–Cordón Caulle southern Chile.

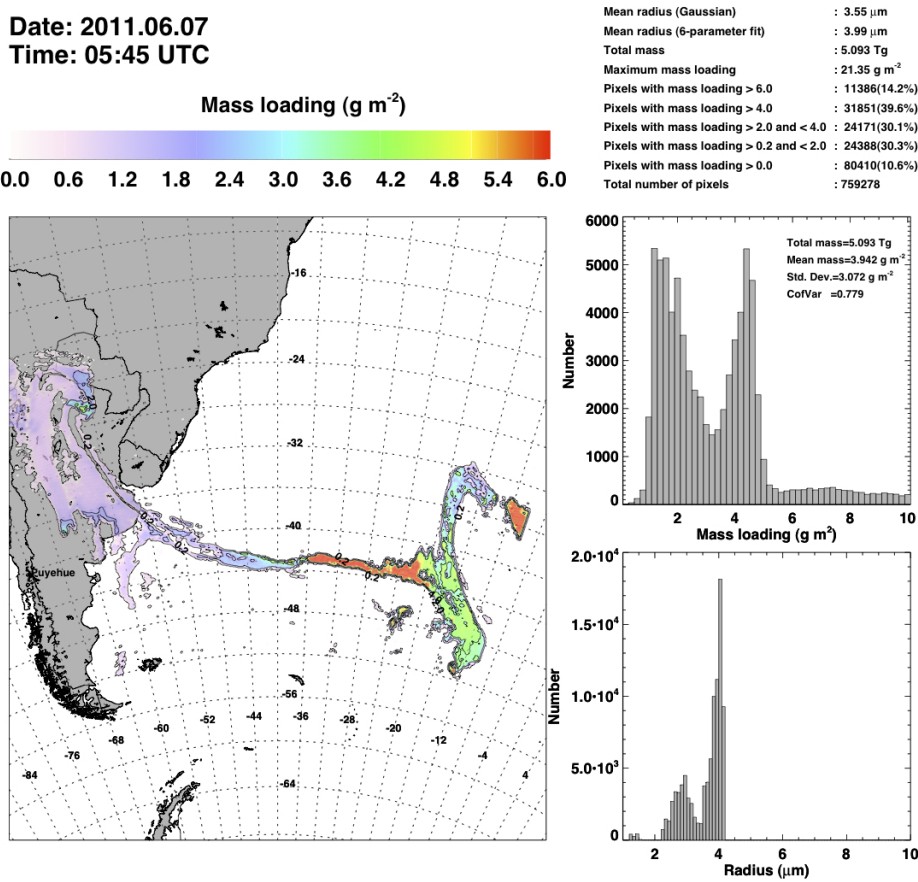

**Figure 6.** SEVIRI ash mass concentrations (g m$^{-2}$) for an eruption of Puyehue–Cordón Caulle volcano, southern Chile in June, 2011.

There are many satellites (polar and geosynchronous) that carry these infrared channels (see Tables 1 and 2), so this product can be delivered globally.

### 4.1. Refractive Index and Composition

Ash clouds contain mixtures of highly fragmented tephra, hot gases and commonly hydrometeors in the form of water droplets and ice. It is also possible to have ice coated ash particles—see later. The mineral composition and internal structure of the ash varies between different eruptions and within the same eruption at different times. Tables 3 and 4 illustrate the mineral types and their abundances from several different volcanoes.

**Table 3.** Composition of ash from some recent volcanic eruptions.

| | Volcano | | | | | | |
|---|---|---|---|---|---|---|---|
| **Oxide** | **Rinjani** | **Agung** | **Cháiten** | **Eyjafjallajökull** | **Grímsvötn** | **Etna** | **Askja** |
| $SiO_2$ | 64.29 | 53.82 | 73.23 | 57.38 | 49.13 | 47.14 | 70.65 |
| $TiO_2$ | 0.58 | 1.06 | 0.15 | 1.52 | 2.84 | 1.76 | 0.84 |
| $Al_2O_3$ | 18.76 | 20.12 | 13.83 | 14.66 | 13.25 | 17.47 | 12.28 |
| $Fe_2O_3$ | 4.41 | 8.75 | 1.60 | 10.02 | 14.87 | 11.38 | 4.35 |
| $FeO$ | 3.58 | 7.10 | – | – | – | – | – |
| $MnO$ | 0.14 | 0.18 | 0.062 | 0.243 | 0.213 | 0.171 | 0.110 |
| $MgO$ | 0.92 | 2.85 | 0.34 | 2.49 | 5.20 | 5.18 | 0.84 |
| $CaO$ | 3.00 | 8.54 | 1.51 | 4.91 | 9.63 | 9.89 | 2.56 |
| $Na_2O$ | 4.15 | 3.32 | 4.18 | 5.53 | 2.82 | 3.60 | 3.96 |
| $K_2O$ | 3.55 | 1.12 | 2.957 | 1.928 | 0.468 | 2.048 | 2.317 |
| $P_2O_5$ | 0.20 | 0.23 | 0.062 | 0.315 | 0.305 | 0.574 | 0.167 |
| $SO_3$ | <0.003 | 0.013 | 0.377 | 0.056 | <0.003 | 0.155 | <0.003 |
| LOI | 5.26 | 2.43 | 1.33 | −0.17 | −0.42 | −0.09 | 1.02 |
| Total | 100.4 | 98.64 | 99.23 | 98.84 | 98.67 | 99.18 | 99.10 |

Ash typically contains $SiO_2$ of >50% by weight. This is important because the main methods for detecting and quantifying ash in volcanic clouds rely on an infrared signature due to $SiO_2$. With knowledge of the $SiO_2$ content or the ratio of non-bridging oxygen to the tetrahedrally-coordinated cations (NBO/T) and the glass and crystal amounts it is possible to select appropriate refractive indices to use for the scattering calculations needed in order to retrieve ash properties from infrared satellite measurements [40].

**Table 4.** As for Table 3, composition of ash from some recent volcanic eruptions.

| | Volcano | | | | | | |
|---|---|---|---|---|---|---|---|
| **Oxide** | **Spurr** | **Redoubt** | **Sakurajima** | **Kelud** | **Merapi** | **Hudson** | **Copahue** |
| $SiO_2$ | 55.42 | 60.45 | 60.0 | 56.1 | 54.69 | 47.60 | 52.07 |
| $TiO_2$ | 0.72 | 0.56 | 0.16 | 0.18 | 0.74 | 2.19 | 1.25 |
| $Al_2O_3$ | 18.76 | 17.83 | 18.3 | 19.2 | 19.29 | 16.35 | 17.54 |
| $Fe_2O_3$ | 7.99 | 6.47 | – | – | – | 11.48 | 8.28 |
| $FeO$ | – | – | 5.70 | 4.89 | 7.76 | – | – |
| $MnO$ | 0.152 | 0.145 | 0.07 | 0.14 | 0.19 | 1.96 | 1.40 |
| $MgO$ | 4.40 | 2.41 | 4.10 | 5.33 | 2.25 | 4.37 | 4.39 |
| $CaO$ | 7.55 | 6.27 | 7.41 | 11.6 | 8.12 | 8.23 | 7.09 |
| $Na_2O$ | 3.44 | 4.01 | 3.27 | 2.26 | 3.73 | 4.08 | 3.60 |
| $K_2O$ | 0.953 | 1.462 | 0.76 | 0.41 | 2.16 | 1.27 | 1.86 |
| $P_2O_5$ | 0.233 | 0.211 | – | – | 0.30 | 0.74 | 0.28 |
| $SO_3$ | – | – | – | – | 0.03 | – | – |
| LOI | 0.51 | 0.29 | – | – | – | −0.31 | 1.15 |
| Total | 100.30 | 100.12 | 99.70 | 100 | 99.28 | 97.95 | 98.91 |

*4.2. Size Distribution*

The size distribution is required for calculations of the scattering parameters needed for radiative transfer calculations that ultimately provide estimates of mass loadings from satellite retrievals. The size distribution is also needed in dispersion models. The size distribution for tephra ejected into the atmosphere spans several orders of magnitude from less then $\sim$1 μm to more than $\sim$1 cm. So-called fine ash has diameter <63 μm and it is this fraction that is dispersed furthest and remains longest in the atmosphere.

Durant et al. [1] compare the size ranges of volcanic particles present in volcanic emissions, which consist of a mixture of gases (e.g., $H_2O$, $CO_2$, $SO_2$), aerosol, a dispersion of small (diameter, d < 30 μm) solid or liquid particles in a gas medium, and silicate ash particles (d $\leq$ 2000 μm).

Measurements of the size, shape and composition of ash particles are usually made from samples collected after the eruption, in the laboratory and not *in situ*.

The term tephra refers to atmospheric fragmented material ejected during explosive volcanic eruptions. Volcanic ash is a subset of tephra and includes silicate particles in the following categories: coarse ash (63 μm < d ≤ 2000 μm) and fine ash (d ≤ 63 μm); the size range of fine ash includes the majority of the size classification for coarse and fine aerosol particles [1]. Volcanic ash includes a dominant glassy component, a "lithic" component, and a "crystal" component of minerals formed in the magma, and these vary from volcano to volcano (see Tables 3 and 4). The important points to note concerning EO ash retrieval methodology are that the size range of interest covers 1 < d < 32 μm and the composition is high in $SiO_2$.

For the purpose of radiative transfer modelling the log-normal size distribution [41,42] is generally used as an input to the Mie scattering calculations. Since there are so few *in situ* airborne measurements of the size distribution, this aspect of the modelling remains very much unconstrained.

### 4.3. Optical Depth

Optical depth is an important concept in radiative transfer and is a parameter that can be directly determined from IR satellite measurements. An alternate method of retrieving ash mass loading and effective particle radius is to pose the radiative transfer problem in a different way. Instead of considering the radiation components of the measurement, it is possible to start with a mathematical expression for the optical depth of the cloud and solve a different inversion problem. The directional optical depth of an arbitrary shaped cloud where the geometrical thickness $z$ is along the viewing direction may be written:

$$t = z \int_0^\infty Q_\lambda(r, \mathbb{R}) \pi r^2 \frac{dn}{dr}(r) dr, \tag{18}$$

where $t$ is the directional optical depth, $\mathbb{R}$ is the complex index of refraction of the cloud particles, $r$ is particle effective radius, $n$ is the particle size distribution, and $Q_\lambda$ is the extinction efficiency. The goal here is to solve for the particle size distribution, or the moments of an assumed distribution. The problem is recognised as a Fredholm integral and the solution, once again, requires the inversion of an underdetermined system of equations. In its general form,

$$G(\lambda) = \int F(r) K(\lambda, r) dr, \tag{19}$$

where $K$ is a kernel function consisting of derivatives of parameters and $F(r)$ is the function to be derived, which in our case is the size distribution. Instruments with many channels, e.g., hyperspectral imagers are best suited to this approach as the size distribution requires many values of the effective radius to be properly determined and this necessarily means many measurements are required.

This new method and the techniques already developed by [22,28,33,34,37,43] are being improved. Further details of these quantitative techniques can be found in the references and a summary of the status of quantitative remote sounding of volcanic ash can be found in [44].

The intention here is not to explore retrieval methods further but rather to look at the problem of ash detection from the perspective of EO satellites and show how multispectral, multiangle and multitemporal resolution instruments they can be used to provide information to reduce ash detection uncertainty.

### 4.4. Setting the Detection Threshold

A key aspect to IR ash detection is setting a brightness temperature difference (BTD) threshold ($DT_c$) below which pixels are classified as containing ash. The theoretical value is $DT_c = 0$ K. However there are a number of reasons why this value may not be accurate and may need to be dynamically adjusted. Some of these reasons are:

- The effect of water vapour absorption, which is highly variable, causes the BTD to increase so that it can be positive for ash affected pixels.
- Variable, spectral emissivity of the underlying surface can cause the BTD to be positive for ash affected pixels,
- Misalignment of the instantaneous fields of view (IFOVs) of the IR channels can cause the BTD to appear smaller or larger than expected, depending on the heterogeneity of the scene.
- Sub-pixel or mixed pixel effects can cause the BTD to appear smaller or larger than otherwise expected, depending on the scene heterogeneity.

It is not possible to provide one simple recipe for setting the threshold because these effects can be present in different amounts for different instruments, different conditions and at different times for the same instrument. A method for indirectly setting thresholds is to use a climatological approach whereby ash affected pixels are detected as "outliers" from what might be typical values for brightness temperatures or reflectances. Such a method has been investigated by Pergola and colleagues [45]. Currently it is recommended that the threshold be set dynamically but that some type of cloud classification or series of cloud tests be used to identify pixels based on more than just the IR threshold. To illustrate the problem of setting a threshold, data from the AATSR have been used to identify ash pixels for ash clouds from the Eyjafjallajökull eruption on 6th May 2010 at 12:41UT. Figure 7 illustrates the effect of changing the threshold value on the number of pixels classified as ash.

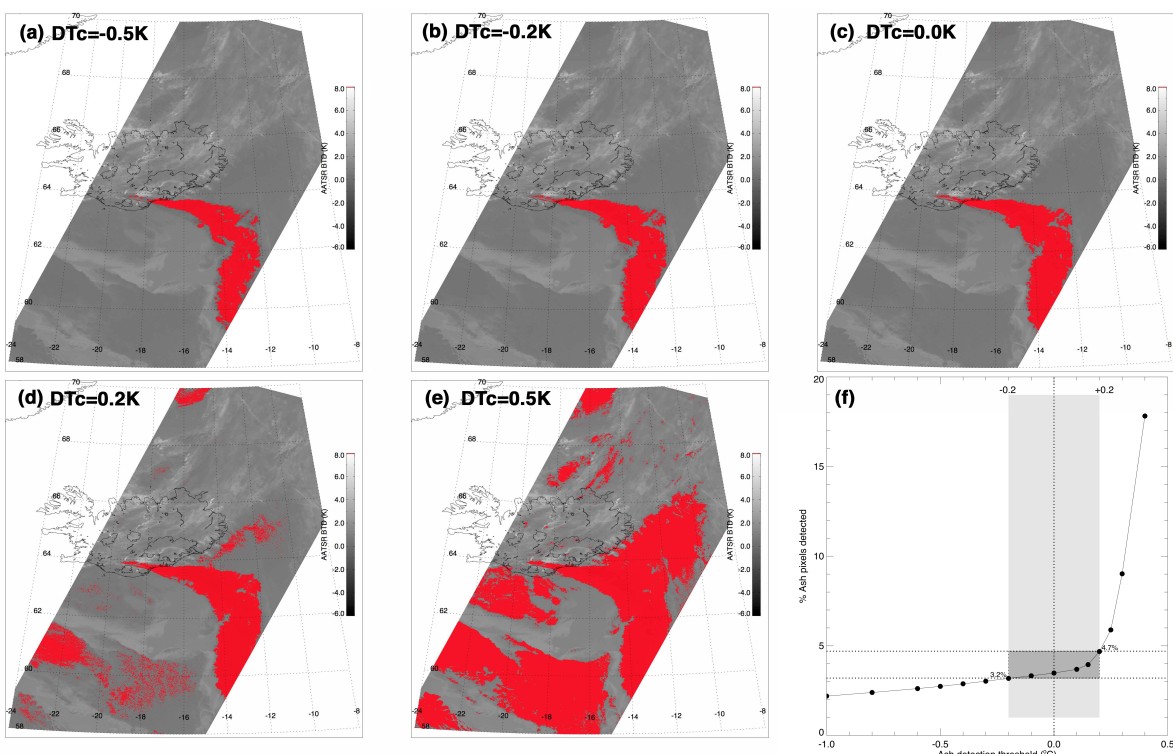

**Figure 7.** Ash detection based on brightness temperature difference thresholds ($DT_c$). (**a**) $DT_c = -0.5$ K, (**b**) $DT_c = -0.2$ K, (**c**) $DT_c = 0.0$ K, (**d**) $DT_c = +0.2$ K, (**e**) $DT_c = +0.5$ K. (**f**) The variation of the number of pixels classified as ash (in %) as a function of $DT_c$.

The number of pixels classified as ash increases rapidly as $DT_c$ increases beyond +0.2 K. The threshold can be objectively set by calculating $DT_c$ at the turning point of the curve.

*4.5. Using More Than Two Channels*

Some imaging IR instruments incorporate several narrowband channels between 6–14 μm and in principle these can be used to determine information on cloud properties. With two channels it is

possible to determine two quantities, viz. optical depth and particle radius. MODIS has channels at 6.7, 7.3, 8.6, 10.8, 11.9 ad 13.2 µm so in principle six properties can be determined. Figure 8 shows MODIS data for six IR channels. Each channel (right-hand boxes) is labelled with its corresponding central channel wavelength and the parameter that most affects the measurement. The left-hand panels show (top-right) the effective radius retrieved assuming an underlying log-normal distribution, and the optical depth ratio ($\beta$) for the 10.8 and 11.9 µm channels (bottom-left). The distribution of the retrieved effective radii is shown in the middle-top panel and the brightness temperature distribution for the 10.8 and 11.9 µm channels is shown in the middle-bottom panel. The red-curves illustrate the parametrised heuristic model with different values for $\beta$. The right-hand IR channel images show a great deal of similarity but also significant differences, suggesting that the brightness temperatures are being affected by different substances, e.g., water molecules, $SO_2$ gas and ash particles. The longest wavelength channel (13.2 µm) is a good indicator of the cloud-top temperature in opaque regions of the cloud.

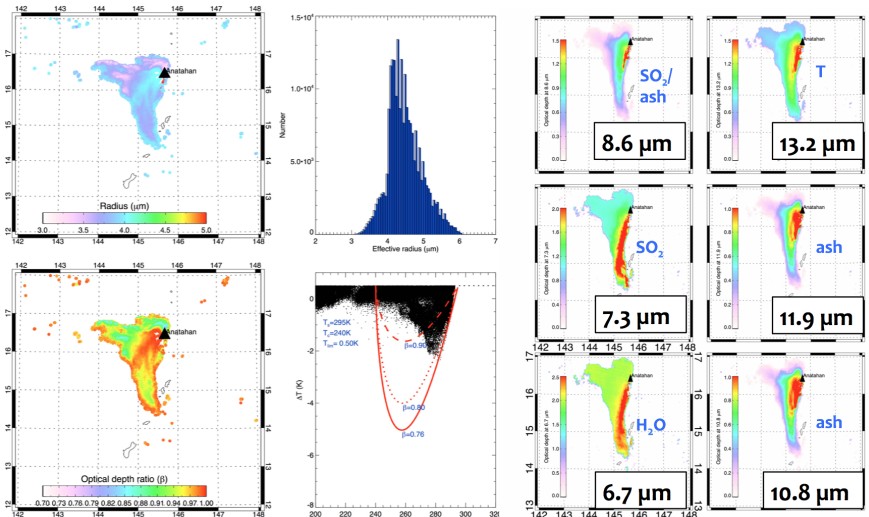

**Figure 8.** Illustration of the information content in each of the IR MODIS channels for a volcanic cloud from the eruption of Anatahan volcano, Northern Mariana Islands in July 2003.

During the day it is also possible to use other (visible and near infrared) channels to help identify ash clouds.

*4.6. Exploiting Angular Dependence*

Extra information used to constrain the inversion problem can come from the spectral domain, as in the case of MODIS or hyperspectral imagers, or in the spatial and time domains. The constraints can also come from angular information, if available. The sea and land surface temperature radiometer (SLSTR) employs a conical scanning mechanism that permits two views of points on the surface from different viewing angles separated in time by two minutes. The SLSTR also has two IR channels with central wavelengths at 10.8 and 11.9 µm thus permitting four near-simultaneous measurements of the same point on the surface. The set of equations to be solved may be written, where the same assumptions have been adopted as for the two channel problem:

$$
\begin{aligned}
r_1(\theta_1) &= e^{-t_{11}} L_1(\theta_1) + (1 - e^{-t_{11}}) B_1[T_c] \\
r_2(\theta_1) &= e^{-t_{12}} L_2(\theta_1) + (1 - e^{-t_{12}}) B_2[T_c] \\
r_3(\theta_2) &= e^{-t_{21}} L_3(\theta_2) + (1 - e^{-t_{21}}) B_3[T_c] \\
r_4(\theta_2) &= e^{-t_{22}} L_4(\theta_2) + (1 - e^{-t_{22}}) B_4[T_c],
\end{aligned}
\tag{20}
$$

where $L_i(\theta_j)$ are the radiances incident at cloud base,

$$t_{ij} = \frac{zk_i}{\cos\theta_j},$$

$$t_{i1} = t_{i2}\frac{\cos\theta_2}{\cos\theta_1}.$$

For SLSTR and its predecessor instruments (ATSR, ATSR-2 and AATSR) the angles $\theta_1$ and $\theta_2$ are close to nadir and 55° with a small variation along the scan. For the ATSR this varies from 52.3–55° in the forward scan and from 0–21.8° in the nadir scan [46]. The ATSR-2 and AATSR are developments from the ATSR and utilise the same scanning geometry but with additional channels. The SLSTR, while based on the same conical scanning principle is somewhat different. The swath width is larger (1420 km in the nadir compared with ~500 km for ATSR, ATSR-2 and AATSR) and some channels have a higher spatial resolution (500 m vs. 1 km). The other main difference is that instead of a forward view, the scan provides a backward view and this is off-set with respect to the nadir view in order to provide overlap with another instrument on board the same platform (Sentinel-3). A summary of the main features of the along-track scanning family of imagers is given in Table 5.

**Table 5.** Main features of the along-track scanning radiometer (ATSR) family of instruments. Operational dates are given as month/year. ● = present; – = absent. [†] Backward view.

| Parameter | ATSR | ATSR-2 | AATSR | SLSTR |
|---|---|---|---|---|
| Channel (width), μm | | | | |
| 0.55 (0.02) | ● | ● | ● | ● |
| 0.67 (0.02) | ● | ● | ● | ● |
| 0.87 (0.02) | ● | ● | ● | ● |
| 1.38 (0.015) | – | – | – | ● |
| 1.61 (0.06) | – | ● | ● | ● |
| 2.25 (0.05) | – | – | – | ● |
| 3.70 (0.38) | ● | ● | ● | ● |
| 10.9 (0.9) | ● | ● | ● | ● |
| 12.0 (1.0) | ● | ● | ● | ● |
| Nadir swath width (km) | 505 | 505 | 505 | 740 |
| Forward swath width (km) | 512 | 512 | 512 | 1420 |
| Nadir angle (centre) (°) | 0 | 0 | 0 | 0 |
| Forward angle (centre (°) | 55 | 55 | 55 | 55 [†] |
| Spatial resolution (km)—SWIR/visible | 1 | 1 | 1 | 0.5 |
| Spatial resolution (km)—Thermal IR | 1 | 1 | 1 | 1 |
| NEΔT @ 300 K (mK) (thermal) | <500 | <500 | <500 | <500 |
| Operational dates | 7/1991–3/2000 | 4/1995–9/2011 | 3/2002–5/2012 | 2/2016–present (S3A) 4/2018–present (S3B) |

There are currently two SLSTRs in orbit on Sentinel-3A and 3B. The purpose of the dual view was to provide extra information to correct the IR data for the effects of atmospheric water vapour absorption and hence derive a more accurate sea surface temperature. An added benefit of the dual view is that it can provide an estimate of cloud top height from the parallax between the nadir and forward (or backward in the case of SLSTR) views [47].

The set of Equations (20) can be solved by introducing values for $T_c$ and $T_s$ as before, and the solution should be more stable because of the extra measurement constraints (four instead of two). The extra information contained in the angular data is illustrated in Figure 9.

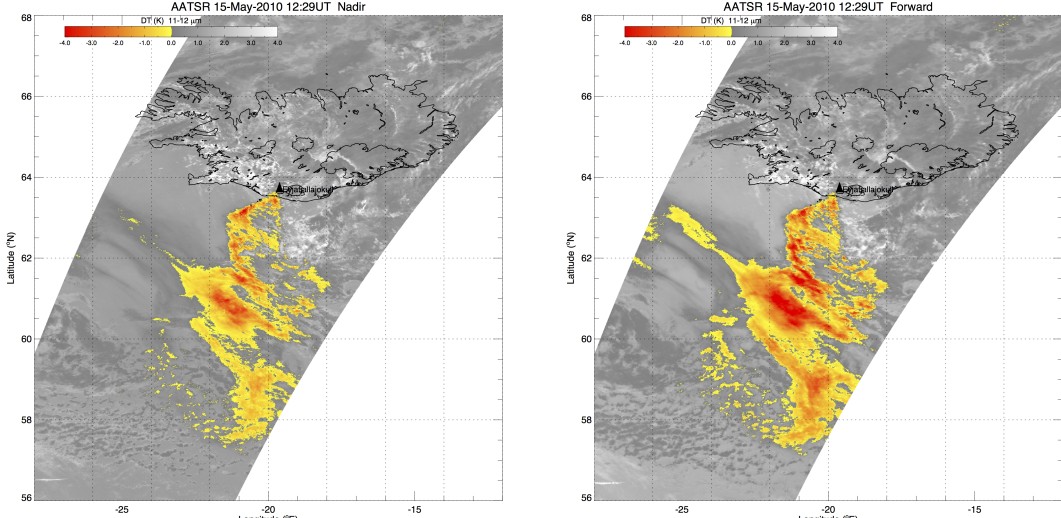

**Figure 9.** (**left-panel**) Nadir AATSR brightness temperature difference (brightness temperature difference BTD=$T_{11} - -T_{12}$ µm) image. Pixels with BTD < 0 K are coloured in shades of yellow to red. Images such as these are extremely useful for assessing the spatial extent and boundaries of ash clouds, but do not provide information on altitude. (**right-panel**) Forward view ($\sim$55°) obtained at the same time as the nadir view and scaled the same way. Note that the spatial extent of the ash is a little larger and the sensitivity is better, due to the more oblique view.

The extra sensitivity of the forward view is mostly due to the extra pathlength traversed by the radiation as it passes through the ash cloud. This effects the amount of absorption, which is proportional to $\exp(-kz)$, where $z$ is the thickness of the cloud in the direction of view. Thus $z$ is increased by a factor of $\sec\theta_f$ or almost a factor of two for the AATSR with $\theta_f \approx 55°$. Figure 9 can be compared with the MODIS/Aqua true-colour image obtained just 45 min earlier (Figure 10).

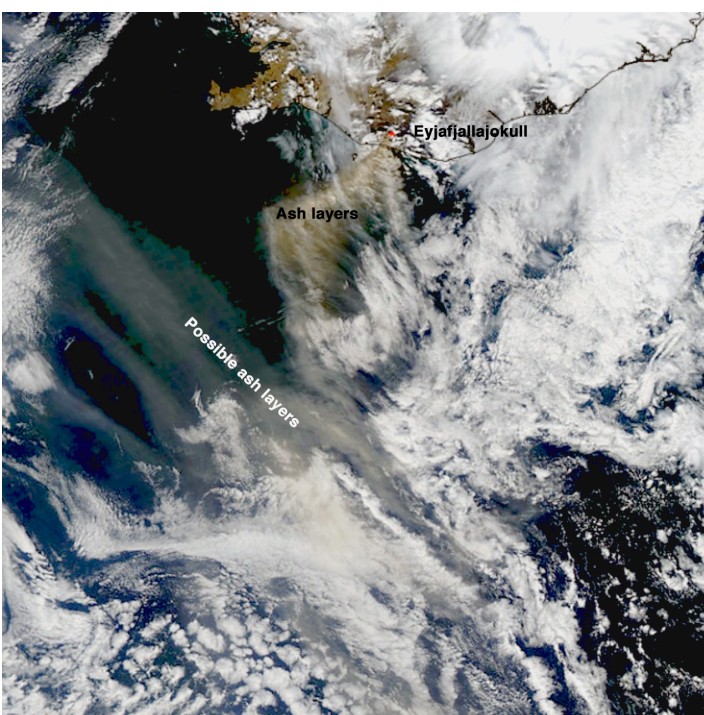

**Figure 10.** MODIS/Aqua true-colour image acquired at 13:40 UT on 15 May 2010. Possible ash layers are indicated on the image.

The extra pathlength of the oblique view is captured well in $T_{11}$ vs. BTD plots as shown in Figure 11.

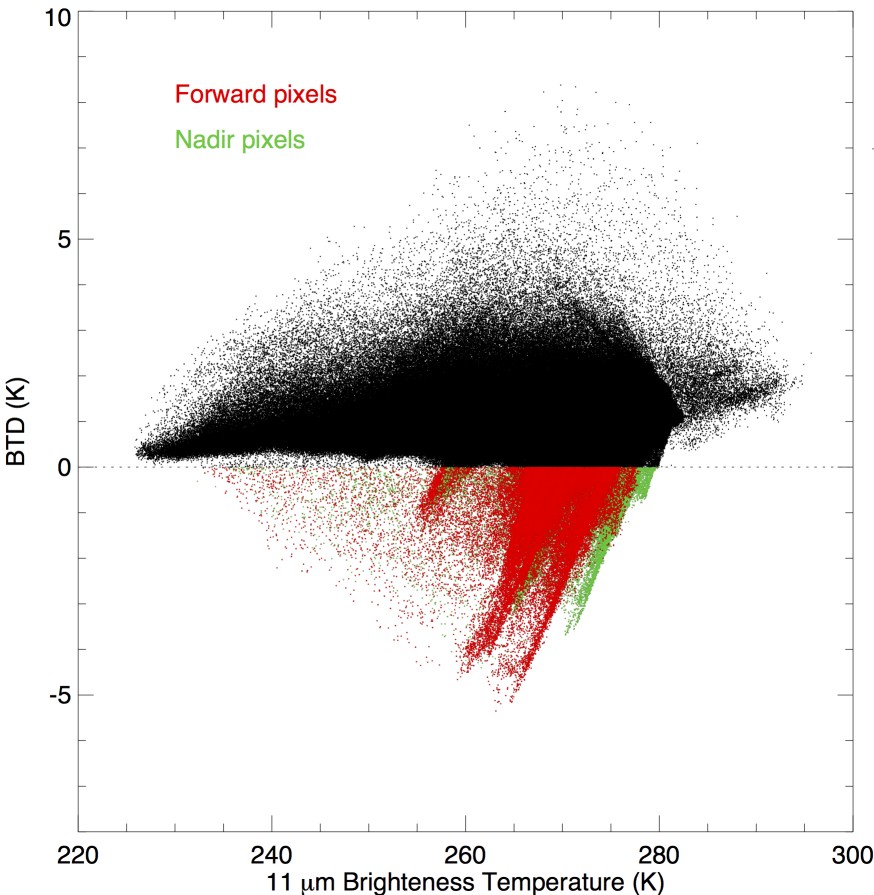

**Figure 11.** Brightness temperature (11 μm) vs. BTD plot for the images shown in Figure 9. Note that for the forward view, the forward 11 μm and 12 μm data have been used. Pixels with BTD < 0 K are indicated in red (forward) and green (nadir).

The forward BTD is more negative than the nadir and more pixels are flagged as "ash" because thin clouds near the detection limit in the nadir appear thicker in the forward view. The 11 μm forward-view brightness temperatures are also shifted to cooler temperatures because of the increased atmospheric absorption.

Altitude information with an accuracy of ±1000 m can be derived from the dual-view capability of the ATSR family of instruments using the methods described in [47,48]. Figure 12 shows a true-colour stereo pair using the near-infrared and visible channels of the SLSTR instrument. A small ash plume from Mt Etna is clearly visible in the images.

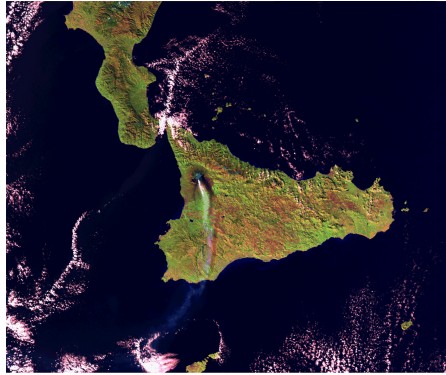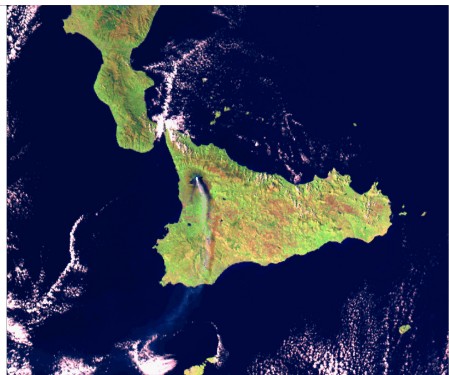

**Figure 12.** Sea and land surface temperature radiometer (SLSTR) stereo pair for an eruption of Mt Etna, Sicily on 27 December 2018 at 09:25 UT. The images are placed so that a third image can be formed by staring at the pair and going slightly "cross-eyed". Elevated clouds will appear to stand out from the image.

By staring at the image pair and third, stereo image, can be made to appear (it is necessary to go "cross-eyed"). Elevated clouds in the image will appear to stand out from the page: the greater the elevation, the greater the effect. Another way to view stereo images like this is the use of red/green glasses and colour enhance the nadir and forward view images appropriately. These qualitative methods of viewing stereo imagery can be very useful for verifying whether certain clouds are high or low and in particular whether cloud layers are above or below each other in multi-cloud scenes (the most common occurrence). For more quantitative assessment the data must be processed and analysed using the methods discussed in [47], for example. Quantitative assessment can be very helpful; for example in determining whether a particular ash layer may be intercepting flight paths and whether it is possible to fly over or under the cloud. During the Eyjafjallajökull ash crisis in April/May 2010 accurate altitude assessment was not available in real-time. To illustrate the utility of AATSR Earth observation for this event, AATSR data acquired on 8 May 2010 at 12:55 UT have been analysed to determine cloud heights. The processing consisted of using the BTD (nadir view) to detect the ash layers (BTD < 0 K) and then using the 1.6 μm near infrared nadir and forward channels to determine heights. Any of the AATSR channels could have been used but the 1.6 μm data provides better correlations when doing the pattern matching (see [47] for further details). The results are shown in the multi-panel Figure 13.

A mean cloud top height of 8±1.5 km is found from the AATSR analysis. There is an indication of higher cloud tops (~11 km) on the southeastern part of the ash cloud band (Figure 13b). The data have been compared to independent height estimates from the Caliop lidar (Figure 13d) and a dispersion model (Figure 13e) and all agree to with ± 1 km. There is both a need and a means for operational cloud top height determination from EO satellites and the current two SLSTRs are ideal candidates for this, because they contain the channels needed for ash detection and the angular scanning necessary for stereo height determination.

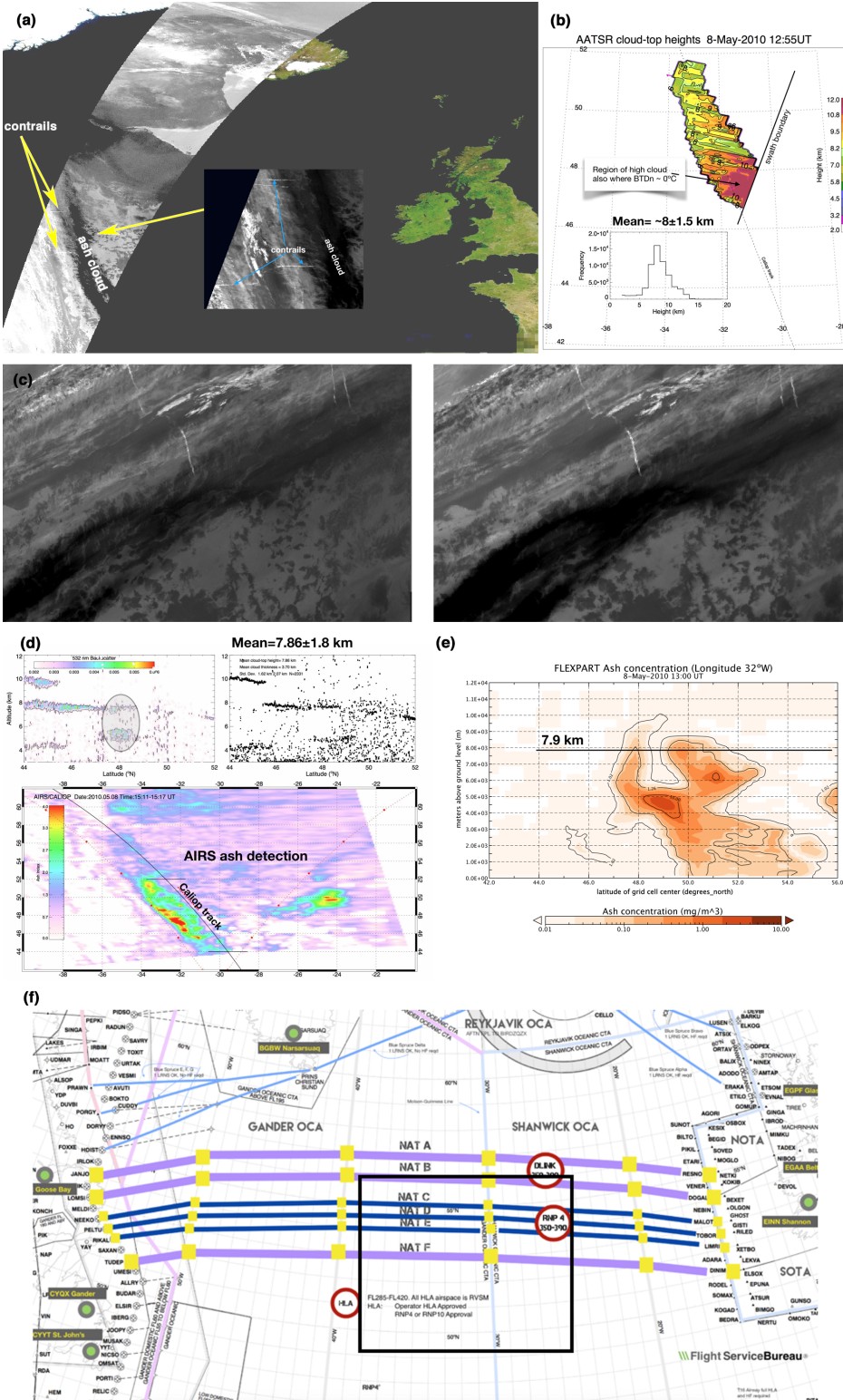

**Figure 13.** AATSR height retrieval. (**a**) BTD image showing location of the AATSR orbit path, ash clouds and contrails. (**b**) AATSR quantitative height retrieval. (**c**) AATSR stereo pair using the 1.6 μm channels. (**d**) Caliop validation data and AIRS ash detection showing the approximate path of the Caliop sub-satellite point. (**e**) FLEXPART modelling for the same ash cloud. (**f**) Location of the north Atlantic tracks (NAT). These are typically at altitudes of 29,000–41,000 ft (~8.8–12.5 km).

## 5. Cloud Identification Scheme (CID)

Identifying or discriminating types of clouds in multispectral imagery is an important process prior to undertaking quantitative retrievals and vital for detecting volcanic ash. There have been many schemes suggested [25,45,49–54] and the details and techniques employed depend on the type of imagery used, especially the number of channels, but also whether there is solar illumination available. As the number of channels increases, the methods tend towards statistically based schemes, principal components, cluster and discriminant analysis. When there are fewer than a dozen or so channels, physically based methods are adopted as the number of independent pieces of information is easier to recognise and the physics is clearer. For example, with five channels, [49] were able to utilise daytime and nighttime tests to identify the fractional amount of cloudiness within an AVHRR pixel.

SEVIRI has 12 channels available for use to identify clouds in imagery; four of these require solar illumination. A detection scheme has been developed based on SEVIRI imagery to assist in the identification of volcanic ash, prior to quantitative retrieval. The main purpose of the scheme is to reduce false detections, but a side benefit is the potential to identify a limited number of meteorological cloud types. The cloud identification scheme (CID) runs a series of tests, mostly using the IR channels. This initial set of tests is based on physical principles (statistical method are also under study) and are categorised into four separate areas that are discussed in more detail below.

### 5.1. Zenith Angle Effects

The SEVIRI field-of-view covers a large area of the Earth's surface encompassing zenith angles of up to $75°$. This also applies to the GOES and Himawari-8 geostationary satellite instruments. At high zenith angles, not only are pixel sizes much large than at nadir, but the viewing effects are rather complicated. This is because at these angles the tops of the clouds are viewed obliquely permitting radiation from the sky behind and the surface below to enter the field of view; in contrast to more vertical viewing where only the cloud top is viewed or the surface below in the case of semi-transparent clouds. The radiative transfer for high zenith angles is thus different to that where the background is the surface or a cloud layer below. Rather than implement more complicated radiative transfer a test for these high zenith angles has been implemented. The test is only applied to pixels which satisfy:

$$T_{11} - T_{12} < \Delta T_0, \text{and } \zeta > \zeta_{max}, \zeta_{max} = 75°, \tag{21}$$

where $\zeta$ is the zenith view angle measured at the surface.

### 5.2. Land Surface Effects

During clear nights over the land surface, spectral emissivity effects can lead to negative BTDs [55]. Extensive research on land surface emissivity [56–59] has shown that emissivity varies with wavelength, and over large areas (the size of a SEVIRI pixel) the large heterogeneity of the surface is lessened so that mean land surface emissivities at 11 μm and 12 μm may be assumed. For vegetated agricultural land in eastern Australia emissivity values of 0.978 (11 μm) and 0.982 (12 μm), while for bare soil in the same region the values were: 0.955 (11 μm) and 0.965 (12 μm) have been found [60]. The observation that the emissivity at 11 μm is always less than the emissivity at 12 μm is relevant, and can be used to correct for clear land effect. Emissivity also varies with zenith ngle [61], and this effect becomes more pronounced at high zenith angles. The emissivity effect over the land disappears during the daytime, mostly because local solar heating of the surface dominates the radiative balance and the emissivity effect is masked. The effects of shadows also becomes more important. In any case the main observation is that on clear nights in the absence of ash clouds, copious numbers of pixels give negative BTDs. To account for this effect the $T_{11}$ and $T_{12}$ brightness temperatures are corrected using the following equations:

$$R_{11\epsilon} = \epsilon_{11} R_{11} + (1 - \epsilon_{11}) R_{11}^a \tag{22}$$

$$R_{12\epsilon} = \epsilon_{12}R_{12} + (1 - \epsilon_{12})R_{12}^a, \tag{23}$$

where $R_{11}$, $R_{12}$, are observed radiances at 11 μm, 12 μm and $R_{11}^a$, $R_{12}^a$ are calculated downwards atmospheric radiances at 11 μm and 12 μm, respectively. The emissivities at 11 μm and 12 μm are $\epsilon_{11}$ and $\epsilon_{12}$, respectively. Brightness temperatures, $T_{11\epsilon}$ and $T_{12\epsilon}$ were then calculated from the radiances using the Planck function. The emissivity correction must be applied to adjust the temperatures as if there was no emissivity effect, which necessitates assumptions on the spectral emissivity in an opposite manner to the known behaviour. In other words the emissivities must be set higher at 11 μm than at 12 μm. The values assumed here were: $\epsilon_{11} = 0.988$, and $\epsilon_{12} = 0.970$. A constant atmospheric temperature of 180 K is used (this probably should be higher but the effect is quite small). The emissivity effect has a diurnal structure so an extra temperature difference, $\Delta T_\epsilon$ is computed as a function of time of day ($t$):

$$\Delta T_\epsilon(t) = \Delta T_o(\cos(\frac{2\pi t}{24}) - 1), \tag{24}$$

where $\Delta T_o$ is a scale factor. The new $T_{11\epsilon}$ and $T_{12\epsilon}$ values are only used over the land.

### 5.3. Cloud Effects

The effects of clouds are many and varied. Several cloud tests have been determined in an ad hoc manner by examining large numbers of images where false-positives are obvious and related to cloud structures. These can most easily be seen when animating imagery. In many cases the clouds appear stationary (e.g., low stratus or fog over ocean is particularly troublesome). Cloud tests rely solely on infrared data (daytime tests are under study) and can use the 3.9 μm channel at night only. Four basic tests have been devised:

- Low cloud uniformity test over the ocean.
- Clouds at moderate to high zenith angles.
- General cloud test.
- Cloud/$SO_2$ test.

These tests are still being refined and make use of the channels at: 3.9 μm, 6.2 μm, 7.3 μm, 8.7 μm, 9.7 μm and 13.2 μm. The 8.7 μm channel is used to identify pixels affected by $SO_2$. The tests were determined by experimentation but have some justification in theory. In their current form, the tests are listed in Table 6 together with the thresholds used. Note that these can change from time to time due to instrument degradation and may not be appropriate to similar channels on other satellite instruments.

### 5.4. Water Vapour

Water vapour acts against ash detection in the sense that high water vapour loadings cause positive BTDs. This effect is well known and has been the basis for estimating sea surface temperatures for over 30 years. A water vapour correction is applied to SEVIRI data based on the method described in [32]. The correction is applied differentially: that is, a larger correction is applied for warmer pixels. The idea follows from the Clausius–Clapeyron relation which indicates that warmer air can carry more water vapour. The effect on the distribution of points in a plot of $T_{11}$ vs. BTD is to rotate the distribution around a point where the correction is 0 K. The water vapour correction is given by Equation (17). An additional water vapour test involves the use of the 6.2 μm and 7.3 μm channels. The 6.2 μm channel is strongly affected by water vapour. The test used is:

$$T_{7.3} - T_{6.2} > \Delta T_6 \text{ and } \Delta T_6 = 20K. \tag{25}$$

### 5.5. Summary and Example of the New Tests

A summary of the tests and the thresholds are provided in Table 6. Note that the thresholds have been determined for just a few cases including Eyjfjallajökull, Grímsvötn and Puyehue-Cordón Caulle and there will be a need to refine and improve these.

The tests cases have enabled some tuning of the thresholds and the detection algorithms. However, this is not complete and more cases are required. Improvements to the CID scheme can certainly be made by including more conditional tests and currently CID does not eliminate false detections due to desert dust outbreaks or resuspended ash events. The results of using the scheme are provided below as histograms for each of the cloud tests on 15 min SEVIRI data for Eyjafjallajökull. A few example images are also provided in order to show the spatial variation of the identifications and for comparison with the traditional brightness temperature difference approach.

On 15 April 2010, ash was first observed streaming from Eyjafjallajökull eastwards towards Norway. At 14:00 UTC a clear ash signal was detected by SEVIRI. Condensed water clouds (possibly also ice rich) are evident in the SEVIRI data. Figures 14–16 show brightness temperatures at 11 μm, 12 μm and 11–12 μm for the SEVIRI frame at 14:00 UTC on 15 April, 2010. These data have been calibrated but not re-projected so they retain true pixel integrity.

In Figure 16 the 'standard' BTD is used to highlight the ash, which in this case is most likely also obscured by water and ice clouds. It is virtually impossible to delineate the ash cloud any better than this using these data alone as cloud above ash or ash embedded in cloud presents the same signature as cloud in the infrared. Perhaps the only way to improve upon this is to utilise a dispersion model simulation and combine that with the ash retrieval. The CID scheme implemented a further 11 tests in attempt to better identify pixels that are ash contaminated, or more aptly, identify pixels that are not ash contaminated. Figure 17 shows the CID scheme for the image of 15 April, 2010 at 14:00 UTC.

The total number of pixels in the image is 759278, and less than 50% of the image is cloud affected; less than 0.5% are identified as ash. Figure 18 provides a summary of the CID pixels as a percent of the total number of pixels. Note that the sum of the percentages of the CID pixels is greater than 100% because multiple tests fail for the same pixel and so these are multiply counted. The most notable observation is that the ash affected pixels is less than 0.5% of the total, while it is down by more than a factor 10 over the BTD method (see Figure 16). Clouds overwhelm the pixel identification (test 11), but note that this is not so obvious in Figure 17 because the colours representing other tests are overlaid onto the same pixels.

Table 6. Infrared cloud tests to identify cloudy pixels in SEVIRI imagery. Test parameters and thresholds have been tuned and may need adjusting from time and time.

| Test | Algorithm | Criteria | Description |
|------|-----------|----------|-------------|
| 0 | $T_{11} - T_{12} < \Delta T_0$ | $\Delta T_0 = -0.8$ K | BTD, reverse absorption Prata (1989b) |
| 1 | $T_{13.2} - T_{9.7} < \Delta T_2$ | $\Delta T_2 = 0.0$ K | Cloud test |
| 2 | $T_{11} - T_{12} < \Delta T_1/\cos(\zeta)$ | $\Delta T_1 = -0.2$ K | Zenith angle ($\zeta$) dependent BTD |
| 3 | $\sigma[T_{11} - T_{12}] > \sigma_{N_s}$ | $N_s = 5$ and $\sigma_{N_s} = -0.9$ K (ocean) $-0.3$ K (land) | Spatial uniformity test |
| 4 | $T_{11\epsilon} - T_{12\epsilon} > \Delta T_\epsilon + \Delta T_\epsilon(t)$ and $T_{11} - T_{12} < \Delta T_0$ $T_{12} > T_{250}; \Delta T_\epsilon(t) = -1 + \cos(2\pi\, t/24)$ | $\Delta T_\epsilon = -0.2; T_{250} = 250$ K $\epsilon_{11} = 0.988, \epsilon_{12} = 0.970; t = $ time in hours | Emissivity test over land |
| 5 | $T_{9.7} - T_{13.2} > T_{240}$ and $T_{11} - T_{12} < \Delta T_0$ | $T_{240} = 240$ K | Low uniform cloud over ocean |
| 6 | $T_{11} - T_{12} < \Delta T_0$ and $T_{39} - T_{12} > \Delta T_3 \cos(\zeta)$ | $\Delta T_3 = 200$ K | Clouds at high zenith angles at night |
| 7 | $T_{11} - T_{12} < \Delta T_7$ and $T_{86} - T_{11} > \Delta T_5$ | $\Delta T_7 = -0.5$ K | $SO_2$/Ash test. Not used currently |
| 8 | $T_{11} - T_{12} < \Delta T_0$ and $\zeta > \zeta_{max}$ | $\zeta_{max} = 75°$ | Excludes pixels beyond zenith angle |
| 9 | $T_{9.7} - T_{13.2} + T_{7.3} - T_{6.2} > \Delta T_4$ and $\zeta > \zeta_0$ | $\Delta T_4 = 7$ K; $\zeta_0 = 72°$ | High zenith cloud test |
| 10 | $T_{8.7} - T_{11} - 2T_{12} < \Delta T_5$ | $\Delta T_5 = -5$ K | Cloud/$SO_2$ test over the ocean |
| 11 | $T_{7.3} - T_{6.2} > \Delta T_6$ | $\Delta T_6 = 20$ K | Water vapour/high altitude $SO_2$ test |

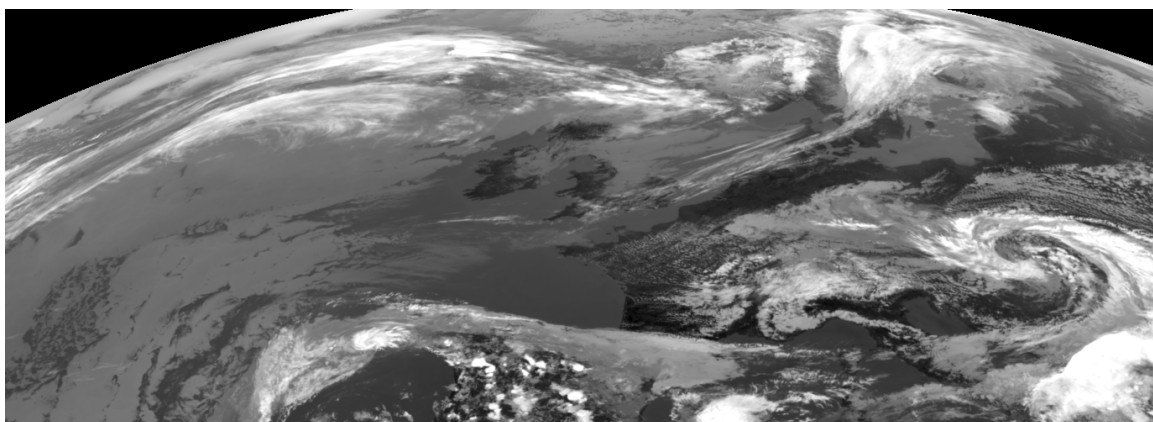

**Figure 14.** Brightness temperature image at 11 μm. The scale ranges from 230 K (white) to 300 K (black).

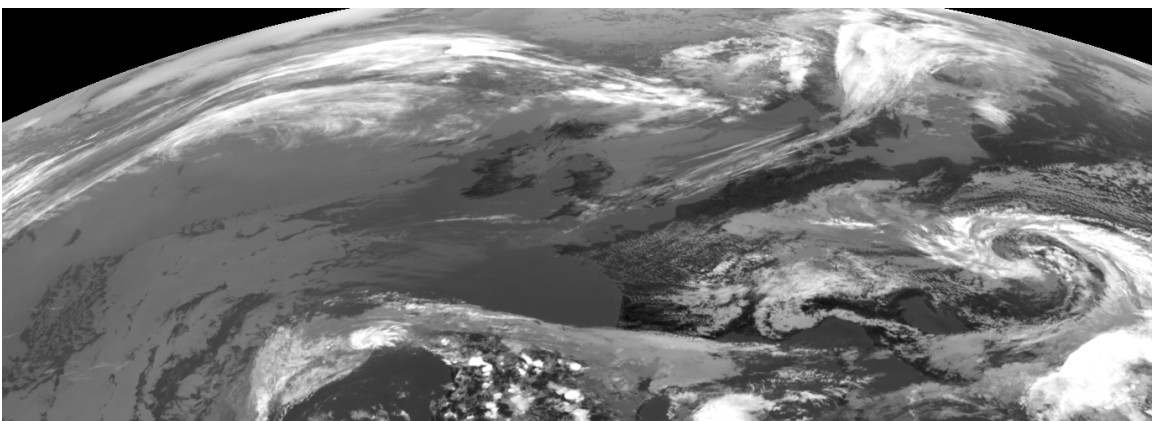

**Figure 15.** Brightness temperature image at 12 μm. The scale ranges from 230 K (white) to 300 K (black).

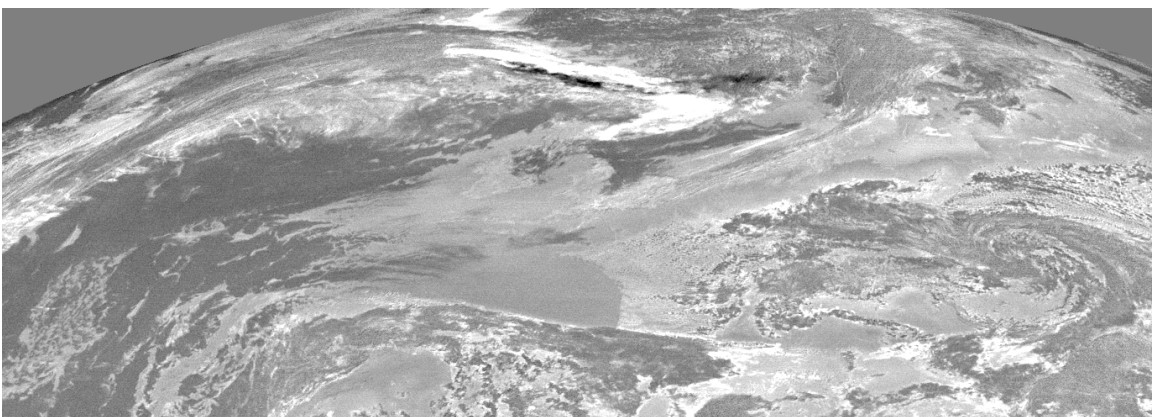

**Figure 16.** Brightness temperature difference image at 11–12 μm. The scale ranges from −3 K (black) to 3 K (white). Ash clouds appear black in these images.

Many cases have been analysed and studied to qualitatively assess the veracity of the new CID scheme but many more cases are still needed.

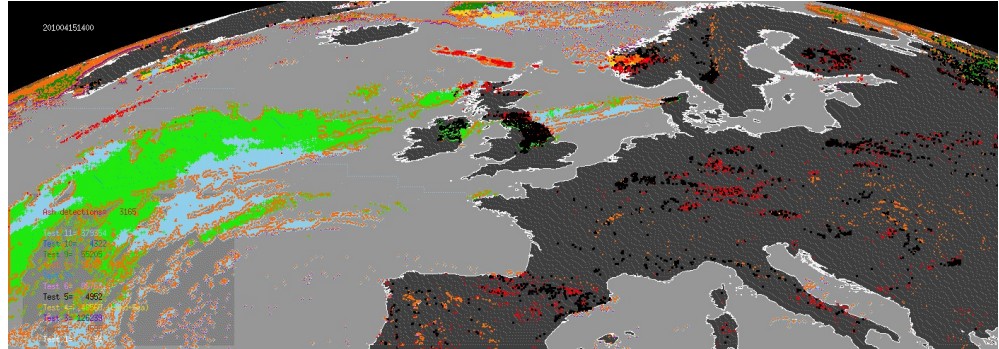

**Figure 17.** Cloud identification scheme (CID) image with colours identifying which tests have been flagged for each pixel. Pixels coloured red are ones that are finally identified as containing ash. Note that some pixels over the land are wrongly identified as ash. The colour legend may be found in Figure 18.

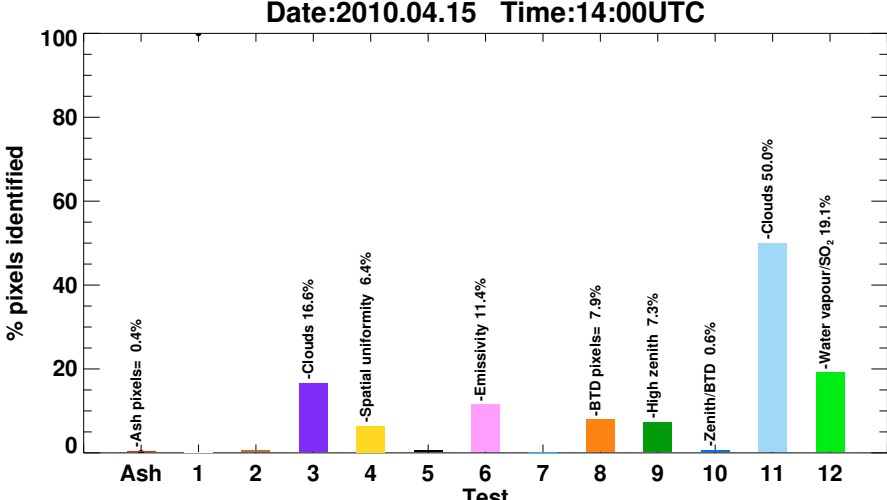

**Figure 18.** Histogram of CID pixels identified by each of the tests. The percentages represent pixels identified by the test out of the total image pixels (759278). The sum of the percentages exceeds 100% because different tests are satisfied by the same pixels.

## 6. Volcanic Ice

Volcanic clouds containing ice particles were first recognised by [22] who showed that IR radiation was extinguished by ice coated ash particles in exactly the same manner as if the particles were composed of pure ice. The giant eruption cloud from the 1994 Rabaul, New Britain volcanic eruption was composed mainly of ice particles [62] and this may be commonplace for phreatomagmatic eruptions. Since then, ice in volcanic clouds have been recognised as an issue for identifying ash clouds [63]. In this discussion, volcanogenic ice is defined as small (radii < 16 μm) ice particles with a solid ash core upon which the ice has nucleated. They are exclusively generated by volcanic eruptions.

One problem of ice formation in erupting volcanic clouds is that the ice "hides" the IR brightness temperature difference signature and makes ash detection and determination of the ash mass loading difficult or impossible. The presence of fine ash in erupting clouds also promotes nucleation of ice, if conditions are favourable [64]. Most importantly there must be sufficient water available, and this can either be entrained from the surrounding atmosphere, as is the case in most tropical eruptions [63], or be present in the eruption column itself, perhaps from sea-water [62]. The process appears to be highly temperature dependent [65] and this suggests that volcanic ash may have only a moderate effect on ice nucleation. The mechanisms leading to ice nucleation by fine ash are complex and not well understood [64–68]. Mineral composition appears also to be an important factor [69,70].

Ice particles in clouds can be detected in the $T_{11}$ vs. BTD curves as the distribution of points in the plot produces an arch, rather than a "U-shaped" distribution. The characteristics of the arch shape depend on the surface temperature, the cloud top temperature, the optical depth and the effective particle radius. Smaller particles generate larger arches, and optically thick clouds tend to suppress the arch. One question that immediately arises is: "can ice particles in non-volcanic clouds be distinguished from volcanogenic ice using the BTD curves?" This question is investigated in two cases studies.

### 6.1. Case study: Sinabung, Indonesia, 19 February 2018

Sinabung volcano (3°10′12″ N, 98°23′31″ E, 2460 m) erupted violently in the morning at 08:53 LT (LT = UT + 7) of 19 February 2018. The column is reported to have reached ~14 km and ashfalls were reported within a region of up to 10 km from the volcano. Satellite imagery from Himawari-8 were acquired and analysed at 10 min intervals covering the period from 01:00 UT (08:00 LT) to 10:00 UT (17:00 LT). The data show a strong ash brightness temperature difference signal from the start of the eruption until 10:00 UT. At ~03:10 UT the eastern edge of the eruption cloud showed signs of ice formation, and an ice-rich cloud continued to form along the eastern and then northern edges. As this ice-rich region evolved, the ash-rich portion of the cloud drifted westwards, with a clear separation of the two parts of the eruption cloud. Separation of constituents in eruption clouds has been noticed and discussed previously [71] in the context of ash and $SO_2$ gas separation for the May 2011 eruption of Grímsvötn in Iceland. Ice and ash separation has not been previously reported in the literature. The separation of the ice-rich portion from the ash-rich portion in this case is due to windshear. Reanalysis wind data show that emissions from Sinabung starting at 02:00 UT on 19 February 2018 travel northwards if at 10 km altitude, whereas emissions at 5 km and 15 km travel northwestwards (see Figure 19).

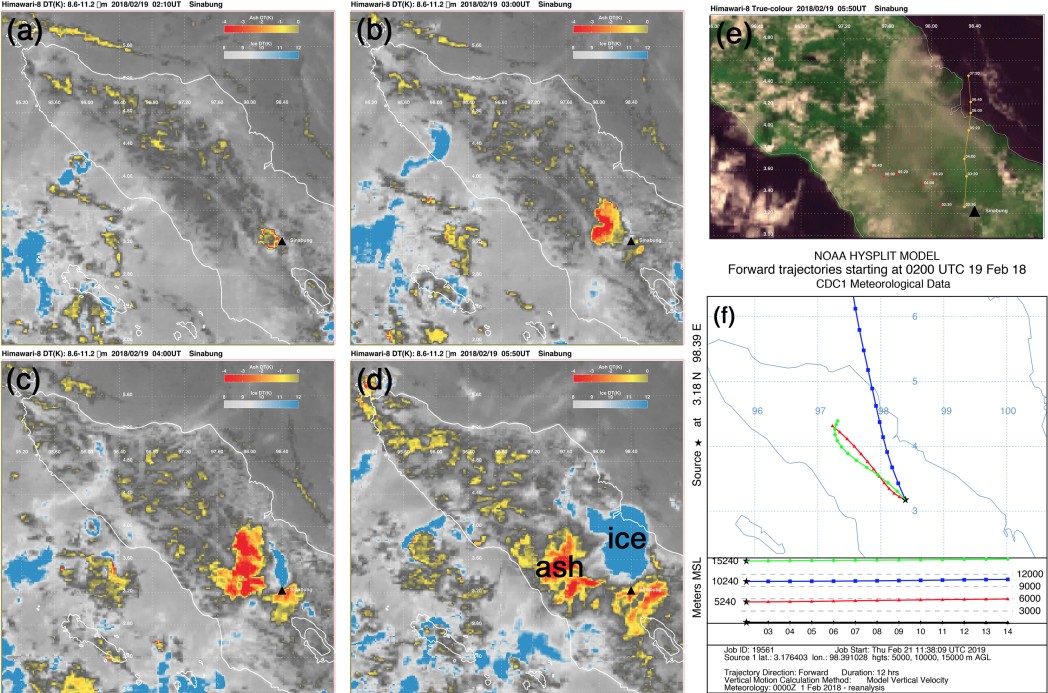

**Figure 19.** Temporal development of the Sinabung eruption cloud. (**a**) Ash signal appears as a ring around an opaque centre. (**b**) Ash cloud grows. (**c**) Ice appears on the eastern edge of the eruption cloud. (**d**) Ice-rich and ash-rich portions separate. (**e**) Trajectories of the most negative (ash) and most positive (ice) brightness temperature pixels at different times. (**f**) HYSPLIT dispersion model trajectories for emissions starting at three different altitudes.

With such clear separation of the ice-rich and ash-rich portions and their different BTD signatures, in this case, it is relatively easy to see the potential for the ice-rich cloud to pose an "ash hazard". The fact that the ice-rich cloud has emerged from within the ash-rich column is highly suggestive that the fine ash has played a role in nucleation. However, the data and analyses cannot prove this. A noticeable aspect of the ice BTD curves is that in general they tend to have much higher arches (strongly positive BTDs) and theoretical modelling suggests this occurs for small particle sizes. Nucleation is known to be more efficient if the surface area of active sites is larger, which is the case for distributions with many small-sized particles.

*6.2. Case Study: Manam, PNG, 31 July 2015*

Manam volcano (4°04′39″ S 145°02′21″ E, 1807 m), a basaltic-andesitic stratovolcano, is situated off the northeastern coast of Papua New Guinea and has a long history (William Dampier sighted several "burning hills", or erupting volcanos off the coast of PNG in his 1699–1700 "Voyage to New Holland" [72]) of volcanic activity. In October/November 2004 Manam had a series of eruptions with column heights reported in the range of 8–18 km. The formation of ice in the 23 November 2004 eruption cloud was investigated [63] using a variety of different satellite instruments, aviation reports, meteorological data and image analysis. Manam is situated in a maritime tropical environment with a plentiful supply of atmospheric water and its larger eruptions are apparently capable of producing volcanogenic ice clouds due to the high water content [63].

At 11:30 LT on 31 July 2015, Manam had a large eruption with a column extending to heights of $>\sim$19 km [73]. Ten minute interval 16 band Himawari-8 data were acquired and analysed for this eruption. Since the eruption occurred in a moist environment, is ash-rich and ice-rich it is a good case to study the veracity of different detection techniques. The data used focused on the early phase of the eruption when the plume was concentrated (opaque in the infrared bands), bright and developing rapidly. Himawari-8 has infrared bands at 7.3, 8.6, 10.2, 11.4, 12.4 and 13.2 µm, of which the middle four bands are known to be useful for ash detection. Various combinations of differences between these bands were used, but in all cases the large opacity of the cloud in the early phase, and the presence of water vapour absorption in the later, more transparent stage of the eruption led to no, or poor detection of ash. As the eruption occurred during the daylight hours it was possible to make full use of the visible and near infrared bands. True-colour imagery strongly suggests the presence of ash based on the brownish colour of parts of the eruption cloud. Of particular interest are the near infrared bands around $\sim$1.6 and 2.2–2.3 µm, which are useful for ice detection [74]. The imaginary part of the refractive index of ice is large at 1.6 µm and small at 2.2–2.3 µm, suggesting that a difference (or ratio) of reflectances in these two channels should be a good indicator if ice in clouds. Figure 20a shows a time-series of the reflectance difference ($\delta\rho = \rho_{2.3} - \rho_{1.6}$ µm) for imagery acquired between 01:20–04:50 UT on 31 July 2015, at two locations: one within the volcanic eruption column, and one outside, within a large ice-rich meteorological cloud some distance upwind of the volcano. The differences show the expected behaviour for the ice cloud: the absorption at 1.6 µm due to ice causes a higher value of $\delta\rho$ compared to water- and ash-rich clouds. Figure 20b shows the time-series of brightness temperature differences (10.2–11.4 µm; $\Delta BT$), where at the start of eruption (01:20–03:20 UT) the high opacity of the cloud gives differences close to zero or slightly positive. After 03:20 UT $\delta\rho$ decreases, while $\Delta BT$ increases. This positive increase was due to a decrease in opacity, allowing the ice (and possibly water vapour) signal to dominate over the ash, by now advected towards the southwest. By comparison, the thick meteorological ice cloud $\Delta BT$ remains close to zero or slightly positive. There was a noticeable decrease in $\delta\rho$ associated with the increase in $\Delta BT$ after 03:20 UT. This was likely due to the opacity decrease, lowering the ice effect on $\delta\rho$. Two images of $\delta\rho$ at 02:00 and 04:00 UT are also shown in Figure 20a to illustrate the change in $\delta\rho$. The locations of the two sites are shown as red dots on these images.

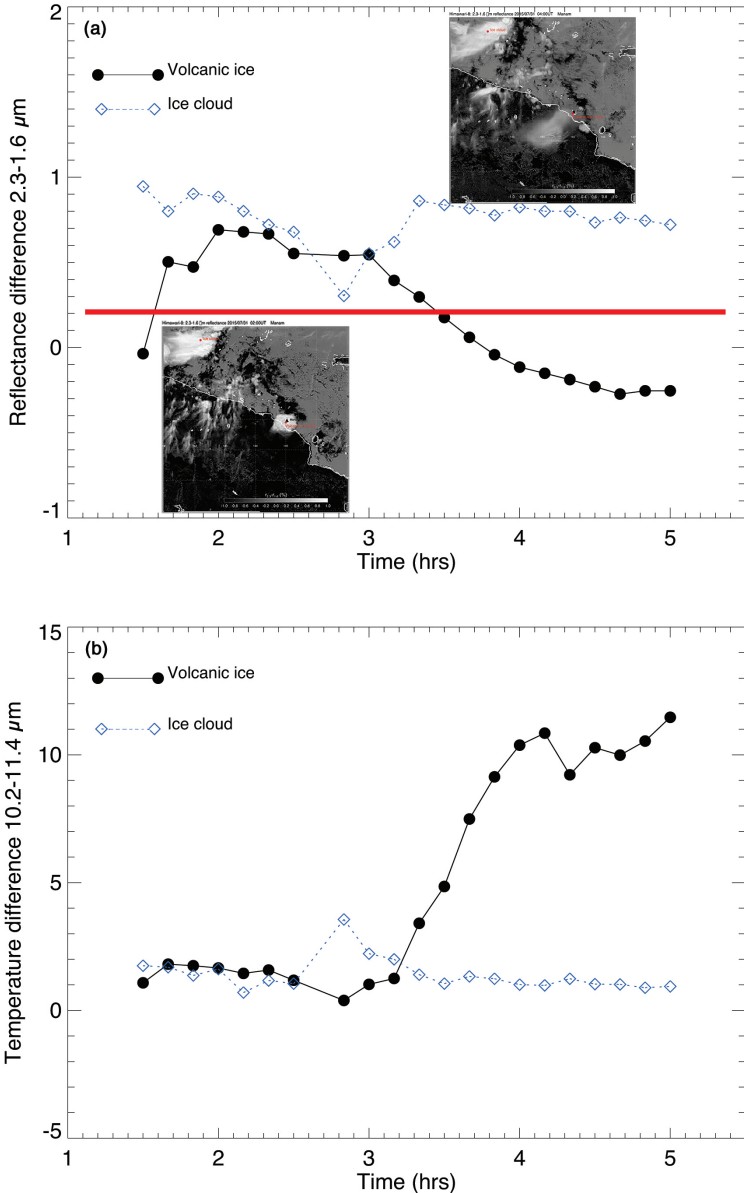

**Figure 20.** Time-series of: (**a**) reflectance differences (2.3–1.6 μm) and (**b**) brightness temperature differences (10.2–11.4 μm), for two locations in Himawari-8 imagery for the 31 July 2015 Manam eruption. The locations were chosen to be close to the erupting column (white line and dots) and near a meteorological ice cloud (blue line and diamonds) upwind from the erupting volcano. The locations of the sites are shown in the 2.3–1.6 μm difference sub-images in (**a**) taken at 02:00 and 04:00 UT.

Using satellite data alone it seems quite difficult to identify volcanic ice from ice formed in meteorological clouds when no ash is present. Detailed models of volcanic plumes [75–77] that include thermodynamics are needed in addition to measurements. Incorporation of meteorological data or using transport and weather forecasting models will assist in diagnosing the circumstances favourable to ice formation in volcanic clouds. Transport models have been used to explore the dispersion of volcanic plumes within a large meteorological system [78].

## 7. Global Volcanic Emission Maps

Volcanic ash is removed from the atmosphere relatively rapidly (within hours to days) but the smallest size fraction (radii < 3 μm) can stay aloft for several weeks if injected high enough into the atmosphere. This was the case during the June 2011 Puyehue–Cordón Caulle (PCC) eruption in southern Chile which sent ash into the upper troposphere and some of the clouds were able to circumnavigate the hemisphere at latitudes south of 40° S. The time to circulate the hemisphere was ≈10 days, suggesting an average zonal velocity of ≈35 ms$^{-1}$. Since PCC erupted many times during the ~21 days of detection by satellite instruments it is not clear whether the same ash circulated more than once. Detection of ash in the atmosphere for more than a few days from a single eruption is unusual; which is not to say that the ash is still aloft and being dispersed but rather that the current satellite instruments lack the sensitivity to detect ash once it falls below a certain concentration (or mass loading). The current sensitivity of passive IR multispectral sensors to detect ash mass loading is ~0.1–0.2 g m$^{-2}$. This is also below the visible threshold for observation by satellite instruments and a typical human eye. Consequently it is difficult to accurately provide "climatologies" of ash residence in the atmosphere. By contrast, $SO_2$ gas is both easier to detect and has a longer residence time. $SO_2$ gas disperses, is removed by wet and dry deposition and converts to $H_2SO_4$ at a rate that depends on the humidity and temperature of the ambient atmosphere in which is resides. The half-life of volcanic $SO_2$ in the atmosphere varies from ~10–28 days. Multispectral passive IR sensors can detect $SO_2$ slant column densities (SCDs) of ~0.01 gm$^{-2}$ so in principle high altitude (Upper Troposphere Lower Stratosphere-UTLS) volcanic $SO_2$ gas can be detected for many weeks. Figure 21 shows a global map of $SO_2$ emissions using AIRS retrievals [79] during 2009. The upper panel shows the times and durations of the emissions, and the lower panel shows the geographic distribution (there is also a list of detected eruptions). Note that AIRS only detects $SO_2$ when it reaches the UTLS, which acts like a filter to mask out lower tropospheric $SO_2$ (mostly from passive degassing volcanoes), leaving only the $SO_2$ detected that might have significance for climate. The spatial distribution is determined largely by the winds and, in general, emissions in the higher latitudes of both hemispheres are confined to their respective hemispheres. Tropical emissions that reach the UTLS are transported zonally with some meridional dispersion and a strong bias to the hemisphere in which the eruption occurred. Transport across the equator into the other hemisphere is not frequent, but possible and depends on the time of year and the position of the inter-tropical convergence zone (ITCZ). Atmospheric dynamics play a key role in the transport of volcanic emissions [78] and while it is not always the case, $SO_2$ dispersion can be used as a proxy for ash transport.

### 7.1. Annual Global Maps

Global composites of AIRS $SO_2$ retrievals for each year from 2002 until 2015 have been compiled to determine the spatial distribution of UTLS $SO_2$ and assess the inter-annual variability. An example is shown in Figure 21 for 2009. The volcanic explosivity index (VEI) [80] is a commonly used measure of the size of an eruption and during 2009 there are 15 erptions with VEI = 2 and just 3 with VEI > 2. Eruptions with VEI = 3 or higher can potentailly inject gases into the UTLS. The eruption of Sarychev Peak in mid-June is the largest eruption of 2009 and the $SO_2$ emissions were quite widespread over the northen hemisphere at latitudes north of 40° N. Figure 22 shows a summary of $SO_2$ emissions determined from AIRS for the years 2002–2015. $SO_2$ masses of more than 1 Tg are rare: the eruption of Kasatochi in August 2008 was the largest emission of $SO_2$ to the UTLS for the whole record. There is no obvious pattern to the emissions other than the frequency of mid-size eruptions (VEI < 3) is most common. The average annual $SO_2$ UTLS emission is 0.5 ± 0.22 Tg.

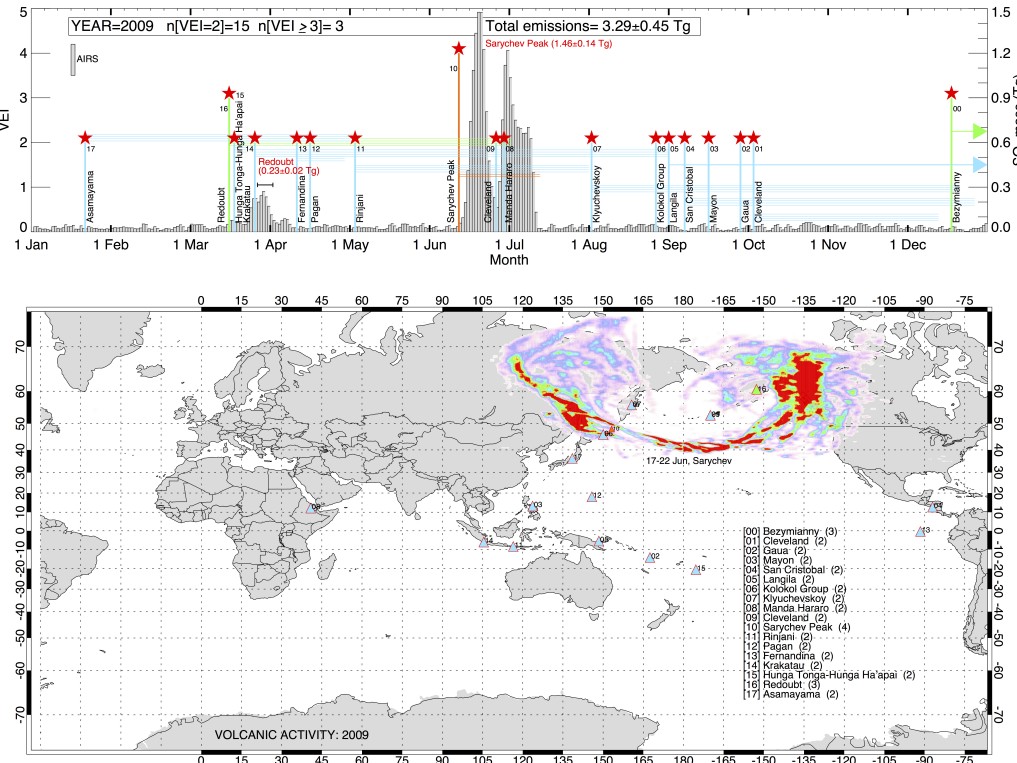

**Figure 21.** SO$_2$ total masses showing volcanic activity during 2009 as detected by the AIRS Earth Observation satellite. Note the strong emissions from Sarychev Peak lasting ∼4 weeks.

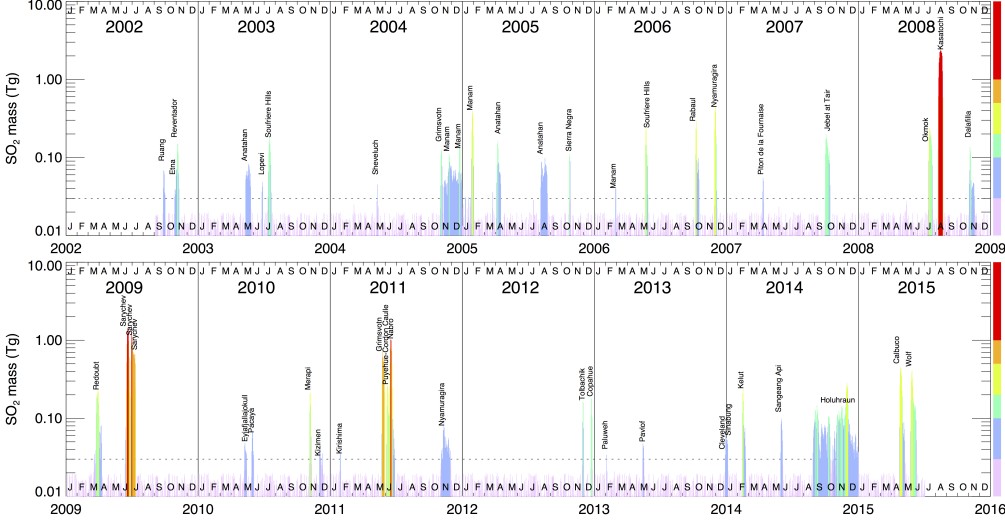

**Figure 22.** SO$_2$ total masses for 2002–2015 derived from AIRS.

### 7.2. Regional Maps

Apart from looking at climatological time-scales of SO$_2$ emissions it is also possible to use satellite instruments to determine daily amounts. As mentioned previously, the role of the atmosphere in dispersing volcanic emissions is critical and SO$_2$ can be used, in some circumstances, as a proxy for the more hazardous ash clouds [81]). For this purpose it is sufficient to simply detect the emissions (a much less arduous task than retrieving quantities) and an index of detected SO$_2$ can be used. Figure 23 illustrates this on 16 August 2008 for SO$_2$ emissions from the eruption of Kasatochi. The filamental nature of the plumes of SO$_2$ is apparent, as is the circumpolar nature of the dispersions with some

meridional transport, especially near and southwards of Iceland. Using wind data it is possible to estimate the height of these $SO_2$ plumes to be ~12 km, which implies that, in some locations, the $SO_2$ has penetrated the tropopause. Verification of the height can be obtained using the Caliop lidar which confirms the plume-top height to be 12 km with a thickness < 1 km (Figure 24). The plume lies within typical cruise altitudes for the North Atlantic tracks (NATs), which vary between 29,000 ft (8.8 km) and 41,000 ft (12.5 km), based on minimisation of tail- and head-winds. The ash mass loading in these plumes is unknown but will be less than the detection limit (~0.2 g m$^{-2}$) for most IR imaging instruments. The $SO_2$ peaks at around 30 DU (Dobson Units). with a mean of about 6 DU along the plume. Taking the mean thickness of the plume to be 0.5 km, the mean $SO_2$ concentration is ~340 µg m$^{-3}$, which is about one half of the recommended exposure level of $SO_2$ averaged over 1 h. The health effects on air passengers is likely to be minimal but exposure of the airframe and aircraft windows to corrosive sulfuric acid may constitute a longer term risk. The use of satellite measurements of $SO_2$, combined with modelling to assess hazards posed by volcanic clouds to aircraft is a growing endeavour and will form part of the safety case and risk assessment that airlines adopt during volcanic crises.

**Figure 23.** AIRS $SO_2$ retrieval for 16 August 2008 for two granules when strong meridional flow is drawing the $SO_2$ plume southwards near to the location of the NATs.

### 7.3. Hemispheric Daily Maps

Hemispheric maps can also be constructed in order to follow the trajectories of ash and $SO_2$ emissions. Winds are generally zonal in the upper atmosphere so the meridional dispersion of the emissions is much smaller than the zonal dispersion. Quite often the $SO_2$ is stretched into long filament structures and while these are impossible to observe in visible light, the infrared signature can be striking. $SO_2$ can generally be tracked for longer periods of time partly because the sensitivity of sensors used to measure $SO_2$ is greater and partly because of the lifetime of ash in the atmosphere is much shorter than that of $SO_2$. Examples of hemispheric maps of both $SO_2$ and ash dispersion in the northern and southern hemispheres, respectively are shown in Figure 25.

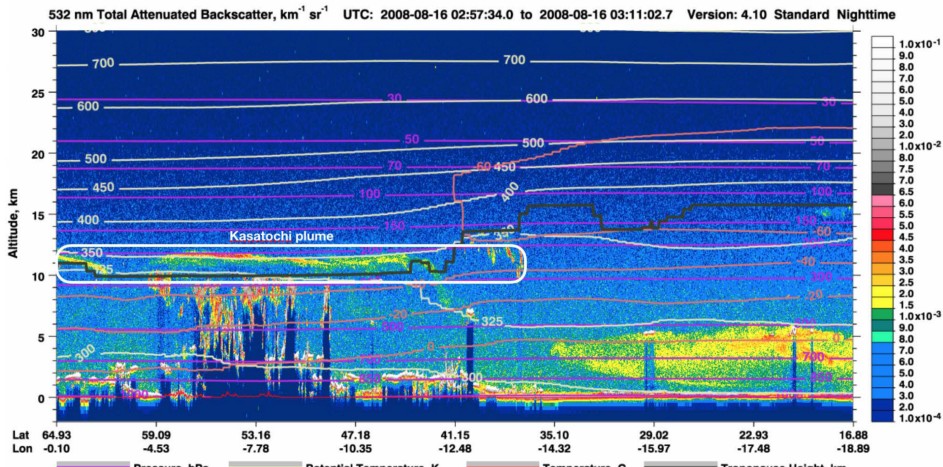

**Figure 24.** Caliop browse image showing location of the Kasatochi plume. The black line indicates the position of the tropopause

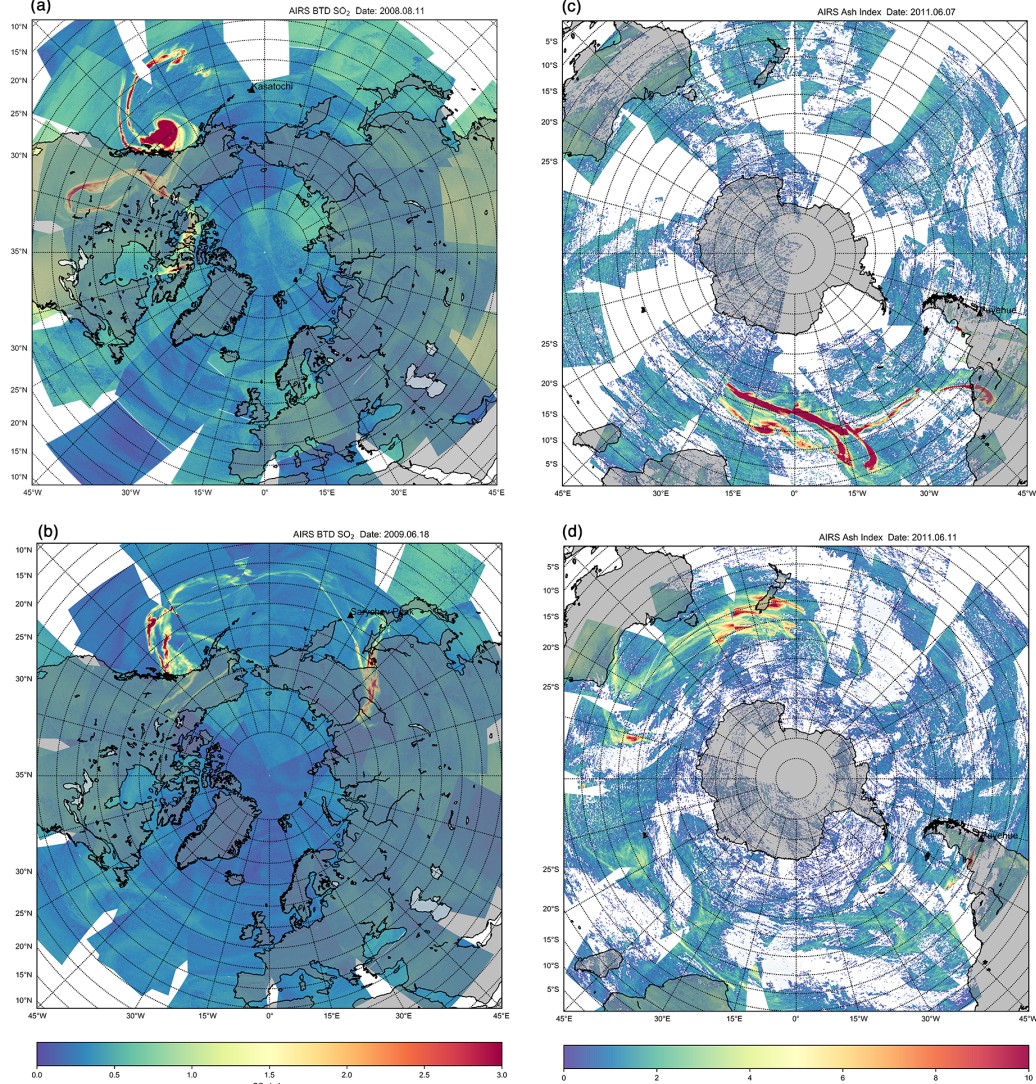

**Figure 25.** Hemispheric maps of the dispersion of $SO_2$ and ash, based on infrared derived indices. (**a**) Kasatochi $SO_2$ 11 August 2008. (**b**) Kasatochi $SO_2$ 16 August 2008. (**c**) Puyehue-Cordón Caulle ash 7 June 2011. (**d**) Puyehue–Cordón Caulle ash 11 June 2011.

With improvements in instrument sensitivity and algorithmic developments it is likely that volcanic emissions will be tracked for longer periods of time and indeed other processes, such as the chemical conversion of $SO_2$ to sulfate in the atmosphere will be monitored. Satellite measurements of volcanic emissions will also be assimilated into global models to improve forecasting of their dispersion and their effects on the atmosphere and climate.

## 8. Conclusions

Earth observing satellite instruments are now well-established tools for monitoring emissions from volcanoes. At this point in time ($\sim$2019), the EO community has a large number of different instruments, both sounding and imaging, on polar and geostationary platforms, that allow quantitative retrievals of important properties of ash and gas clouds. While none of the instruments have been specifically designed to measure volcanic emissions, 30 years of research has seen the development of innovative and in some cases operational methods to assist in the forecasting the dispersion of volcanic clouds, which ultimately helps with aviation hazard mitigation. Quantitative retrieval of $SO_2$ volcanic column abundances from UV and IR instruments provides data for investigating the effects of these emissions on the atmosphere and on its radiative balance. There are sufficient EO instruments and methods available that a large $SO_2$-rich eruption, capable of affecting the climate will be detected rapidly, continuously and quantified so that climate models can assimilate the data and be validated [82].

Some of the issues with detection of volcanic ash using infrared measurements have been discussed and in particular the confounding effects of water vapour, cloud identification and the presence of ice were highlighted. Setting the threshold for ash detection is problematic and it is recommended that no fixed threshold be used, rather a dynamic threshold which depends on image scene character is more appropriate. Computer methods including statistical measures (e.g., PCA or discriminant analysis), neural networks and deep learning algorithms are well-suited to this task. The problem of identifying cloud type was also discussed and a cloud identification (CID) scheme presented. CID is based entirely on multispectral IR data but could be expanded to include visible bands in the future. An example of the scheme illustrates its capabilities and deficiencies.

Estimating the altitude of volcanic clouds is an important and challenging problem from the perspective of IR imaging. The use of stereoscopic IR data was highlighted because of the availability of new sensors in polar orbit that provide two views and permit height determination with $\pm 1$ km. The two Sentinel-3 SLSTR instruments have a heritage going back to the early 1990s with the launch of the first ATSR, designed princiaplly for sea surface temperature measurements. These under-utilised instruments have IR and visible bands with stereo capability and can therefore provide volcanic ash mass loading and height information. The oblique view also provides a degree of extra sensitivity for low concentration ash plumes because of the extra pathlength.

A new class of volcanically-derived particle has been suggested and some examples of its presence in volcanic clouds presented. Volcanogenic ice, a small-sized (radii $\sim< 16$ μm) ice-coated ash particle appears to be common in tropical eruption clouds that reach high altitudes (at least the freezing level). These particles have very strong positive *BTD*s in IR satellite imagery and have a similar signature to that of ice in non-volcanic meteorological clouds. In some cases it is found that volcanogenic-ice *BTD*s exceed those due to ice in non-volcanic meteorological clouds. For ash-rich and ice-rich volcanic clouds of high IR optical depth the *BTD*s are uniformly low or close to zero, eliminating the possibility of using this measure to discriminate between them. Instead it is shown that a near infrared reflectance difference metric: $\delta\rho$ (2.3–1.6 μm reflectance) might be capable of detecting ice, especially when the optical depth is large. The physical principle behind this approach is that absorption due to ice is larger at 1.6 μm compared to that at 2.3 μm. As optical depth decreases, the $\Delta BT$ metric becomes more sensitive, but problems of discrimination between meteorological ice, volcanogenic ice and ash-rich clouds remain.

The case study for the volcanogenic ice and ash from the eruption of Sinabung, Indonesia illustrates a new feature of complex mixed-constituent volcanic clouds: separation of ice-rich and ash-rich portions of the eruption clouds. Previous work [71,83] has shown that volcanic $SO_2$ and ash can travel together or undergo maximum separation, and separation of ice and ash clouds in the 2005 Manam eruption can be seen in [63] but the process is not elaborated. The separation of ice-rich and ash-rich portions of volcanic clouds and indeed of $SO_2$-rich portions, has important consequences for dispersion modelling and forecasting volcanic hazards to aviation. The process also questions the use of a single constituent source term [84], where ash-nucleating ice clouds is occurring *in situ* and the movement is governed entirely by the atmospheric wind structure and in particular on wind shear. While not studied here, preatomagmatic eruptions may be capable of generating copious amounts of volcanogenic-ice clouds [62], thus suggesting source term definitions must include hydrometeors and the complex processes governing hydrometeor phase transitions.

Several EO instruments are particularly well-adapted to measuring $SO_2$ in the atmosphere. The importance of volcanically-derived $SO_2$ to the stratosphere has been recently been reviewed [82]. Some instruments have been able to measure $SO_2$ for three decades or more [85,86], while more recent hyperspectral IR instruments can now provide some vertical height information [30]. Global maps of $SO_2$ from the hyperspectral AIRS instrument show the possibility of delivering climatologies of volcanic-derived $SO_2$. The UV instruments, dating back to the total ozone mapping spectrometer (TOMS) (late 1970s) have now been improved to give unprecedented spatial resolution from the Tropomi sensor [18] and continuous daylight sensing from the EPIC/DSCOVR sensor [6,7].

The focus in this paper has mostly been on passive, near-nadir remote sensing and sounding. There are several instruments that use limb sounding techniques to infer volcanic ash and gas properties [38,39] and these offer improved vertical resolution as well as better sensitivity in some cases. Imaging (or sounding) towards the limb is unavoidable for detection systems mounted on board aircraft [87]. Active systems using radar [88] or lidar [89] provide particle size and altitude information, difficult or impossible to infer from passive infrared data. These topics are too complex and broad to include in our overview.

In summary, the exploitation of passive EO satellite instruments for monitoring and quantifying volcanic emissions is still a growing endeavour but great progress has been made since the early observations of volcanic clouds pioneered by the work of Sawada [90,91] in Japan, Matson [92,93] and Malingreau [94] in the US and Bureau of Meteorology scientists in Australia [21,95,96]. Early summaries of the use of remote sensing data for volcano monitoring can be found in Gupta et al. [97] and Oppenheimer and Rothery [98], while some recent books [99–101] on volcanic ash include several chapters on the quantitative use of EO for volcanic ash and gas quantification.

**Author Contributions:** The majority of the research reported in this paper was conducted by F.P. over many years, beginning in 1986 when he was a post-doctoral research scientist working with M.L. at Curtin University. The authors contributed equally to composing, writing, and revising the paper.

**Funding:** No research funds were provided for this paper.

**Acknowledgments:** The authors gratefully acknowledge the providers of the satellite data used in this work. These include NASA, NOAA, Eumetsat, JAXA/JMA, Copernicus/ESA and ISRO. The authors gratefully acknowledge the NOAA Air Resources Laboratory (ARL) for the provision of the HYSPLIT transport and dispersion model and/or READY website (http://www.ready.noaa.gov) used in this publication. Finally we appreciate the comments by two anonymous referees, whose helpful remarks have improved our paper.

**Conflicts of Interest:** The authors declare no conflicts of interest.

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
