# Peer review of "Passive Earth Observations of Volcanic Clouds in the Atmosphere"

_atmosphere, doi:10.3390/atmos10040199_

Round 1
Reviewer 1 Report
Review of `Earth Observations of Volcanic Clouds in the Atmosphere' by Fred Prata and Mervyn Lynch
General comments
This new paper by Prata and Lynch provides an excellent overview and review of various methods for detection of volcanic clouds in remote sensing observations. The particular focus of this study is on the analysis of geostationary and polar orbiting infrared instruments. A class of `volcanic ice clouds' (referring to ice-coated ash particles) is introduced based and discussed based on two case studies of volcanic eruptions.
Overall, the paper is very well written, clear, and concise. I recommend it for publication subject to few minor comments and fixes as listed below.
Specific comments
p3, l6: Perhaps clarify that the orbits are _nearly_ polar in most cases?
Table 1: The MetOp satellites (IASI) might be added here?
p12, l19: Another infrared radiative transfer model for the analysis of limb sounding measurements of volcanic ash was introduced and applied by Griessbach et al. (2013, 2014).
Griessbach, S., Hoffmann, L., Höpfner, M., Riese, M., & Spang, R. (2013). Scattering in infrared radiative transfer: A comparison between the spectrally averaging model JURASSIC and the line-by-line model KOPRA. Journal of Quantitative Spectroscopy and Radiative Transfer, 127, 102-118.
Griessbach, S., Hoffmann, L., Spang, R., and Riese, M.: Volcanic ash detection with infrared limb sounding: MIPAS observations and radiative transfer simulations, Atmos. Meas. Tech., 7, 1487-1507, https://doi.org/10.5194/amt-7-1487-2014, 2014.
p16, l19-36: Another approach to set detection thresholds might be to conduct a statistical analysis of a long-term record of observations, considering the ash detections as "outliers" in the record?
p23, l17: I got a bit confused about this notation "T_11 - - T_12". Does this mean "T_11 + T_12"?
Fig. 13: The stereoscopic view is quite difficult to get. I didn't manage, I am afraid. However, I like the idea of presenting it.
p24, l28: Does the discrimination between volcanic ash and desert dust also pose a challenge for this scheme?
p29, l30: It seems there is a quite a lot of potential for applying Lagrangian transport simulations in order to investigate the formation/conversion from ash to `volcanic ice'? Perhaps this could be pointed out more clearly?
p32, l27: Wu et al. (2017) performed a comprehensive analyses of the atmospheric transport processes associated with the Sarychev eruption using satellite observations and Lagrangian transport simulations.
Wu, X., Griessbach, S., and Hoffmann, L.: Equatorward dispersion of a high-latitude volcanic plume and its relation to the Asian summer monsoon: a case study of the Sarychev eruption in 2009, Atmos. Chem. Phys., 17, 13439-13455, https://doi.org/10.5194/acp-17-13439-2017, 2017.
p37, l5: Although the focus of this review is on nadir sounding observations, it might be good to refer also to the potential of limb sounding observations (e.g., Griessbach et al., 2013, 2014) for completeness.
Technical corrections
p5, l26-27: fix sentence "Suffice to say..."
p7, l33: fix reference to Rodgers (2000)
p15, l6: < 63 µ_m_
p17, l29: views of a point__
p21, l44: fix "the the"
p23, l11: fix "slghtly"
p24, l39: isolated subsection title 5.5.1 maybe not needed?
p28, l21: multiple tests __ failed
p30, l16: moist environment _and_ was (?)
p34, l7: fix "SO2_2"
p35, l40: was also discussed _and_ the cloud identification
p35, l43: fix "setereoscopic"
Author Response
Reviewer 2
General comments
This new paper by Prata and Lynch provides an excellent overview and
review of various methods for detection of volcanic clouds in remote
sensing observations. The particular focus of this study is on the
analysis of geostationary and polar orbiting infrared instruments. A
class of `volcanic ice clouds' (referring to ice-coated ash particles)
is introduced based and discussed based on two case studies of volcanic
eruptions.
Overall, the paper is very well written, clear, and concise. I recommend
it for publication subject to few minor comments and fixes as listed
below.
Specific comments
p3, l6: Perhaps clarify that the orbits are _nearly_ polar in most
cases?
Yes. We have added the word “near” in the caption to the Table.
Table 1: The MetOp satellites (IASI) might be added here?
Yes. An entry with the MetOp satellites has been added to the Table.
p12, l19: Another infrared radiative transfer model for the analysis of
limb sounding measurements of volcanic ash was introduced and applied by
Griessbach et al. (2013, 2014).
Thanks. We have added both references.
Griessbach, S., Hoffmann, L., Höpfner, M., Riese, M., & Spang, R.
(2013). Scattering in infrared radiative transfer: A comparison between
the spectrally averaging model JURASSIC and the line-by-line model
KOPRA. Journal of Quantitative Spectroscopy and Radiative Transfer, 127,
102-118.
Griessbach, S., Hoffmann, L., Spang, R., and Riese, M.: Volcanic ash
detection with infrared limb sounding: MIPAS observations and radiative
transfer simulations, Atmos. Meas. Tech., 7, 1487-1507,
https://doi.org/10.5194/amt-7-1487-2014, 2014.
p16, l19-36: Another approach to set detection thresholds might be to
conduct a statistical analysis of a long-term record of observations,
considering the ash detections as "outliers" in the record?
This is a very good point and we have added a sentence and included a new reference.
p23, l17: I got a bit confused about this notation "T_11 - - T_12". Does
this mean "T_11 + T_12"?
Corrected. This was supposed to be a minus sign.
Fig. 13: The stereoscopic view is quite difficult to get. I didn't
manage, I am afraid. However, I like the idea of presenting it.
It is not always possible to get the stereoscopic effect but our study and the papers quoted show some promise.
p24, l28: Does the discrimination between volcanic ash and desert dust
also pose a challenge for this scheme?
No. Dust is still a problem. We have added a sentence to indicate this.
p29, l30: It seems there is a quite a lot of potential for applying
Lagrangian transport simulations in order to investigate the
formation/conversion from ash to `volcanic ice'? Perhaps this could be
pointed out more clearly?
Thanks. We have added a new sentence and several references, including the reference suggested.
p32, l27: Wu et al. (2017) performed a comprehensive analyses of the
atmospheric transport processes associated with the Sarychev eruption
using satellite observations and Lagrangian transport simulations.
Paper now cited and a short sentence added.
Wu, X., Griessbach, S., and Hoffmann, L.: Equatorward dispersion of a
high-latitude volcanic plume and its relation to the Asian summer
monsoon: a case study of the Sarychev eruption in 2009, Atmos. Chem.
Phys., 17, 13439-13455, https://doi.org/10.5194/acp-17-13439-2017, 2017.
p37, l5: Although the focus of this review is on nadir sounding
observations, it might be good to refer also to the potential of limb
sounding observations (e.g., Griessbach et al., 2013, 2014) for
completeness.
This is true and we should have included something on this. To rectify the omission we have added a paragraph in the Conclusions and also referred to active sensing. It is not possible to do justice to these topics without greatly expanding the paper.
Technical corrections
p5, l26-27: fix sentence "Suffice to say..."
Fixed.
p7, l33: fix reference to Rodgers (2000)
Proper citation used.
p15, l6: < 63 µ_m_
Change to µm.
p17, l29: views of a point__
Changed to “points”
p21, l44: fix "the the"
Deleted “the”.
p23, l11: fix "slghtly"
Spelling corrected.
p24, l39: isolated subsection title 5.5.1 maybe not needed?
Removed.
p28, l21: multiple tests __ failed
Corrected.
p30, l16: moist environment _and_ was (?)
Grammar corrected.
p34, l7: fix "SO2_2"
Deleted first “2”
p35, l40: was also discussed _and_ the cloud identification
Grammar corrected.
p35, l43: fix "setereoscopic"
Spelling changed to “stereoscopic”
---------------------------------------------------------------------------------------------------
Dr Fred Prata and Prof. Mervyn Lynch
24 March 2019.
Reviewer 2 Report
This is an excellent review paper, that is a model of clarity. It presents an accessible and critical account of the approaches used for Earth Observation of volcanic clouds (in the most general sense), and does an excellent job of both explaining the current state of the art, and identifying knowledge gaps and new opportunities for research. It is written economically, and is in very good shape. The only suggestion, for completeness, that I would encourage the authors to consider would be to cite data sources in the captions to tables and figures (it may well be that the data sources are already cited elsewhere in the text, in most cases).
Author Response
This is an excellent review paper, that is a model of clarity. It
presents an accessible and critical account of the approaches used for
Earth Observation of volcanic clouds (in the most general sense), and
does an excellent job of both explaining the current state of the art,
and identifying knowledge gaps and new opportunities for research. It is
written economically, and is in very good shape. The only suggestion,
for completeness, that I would encourage the authors to consider would
be to cite data sources in the captions to tables and figures (it may
well be that the data sources are already cited elsewhere in the text,
in most cases).
Thank you for the comments. It was remiss of us not to cite the data sources. Rather than add these to every Figure and Table (where needed) we have included an acknowledgement to all of the relevant data providers. We have also acknowledged the HYSPLIT model provider and the two anonymous referees.